# Accelerating multielectron reduction at Cu$_x$O nanograins interfaces with controlled local electric field

Weihua Guo[1,2,10], Siwei Zhang[3,10], Junjie Zhang[4], Haoran Wu[5], Yangbo Ma[1], Yun Song[1], Le Cheng[1], Liang Chang[6], Geng Li[1], Yong Liu[1], Guodan Wei ®[6], Lin Gan ®[6], Minghui Zhu ®[5] ✉, Shibo Xi[7] ✉, Xue Wang ®[8], Boris I. Yakobson ®[4] ✉, Ben Zhong Tang ®[3,9] ✉ & Ruquan Ye ®[1,2] ✉

Regulating electron transport rate and ion concentrations in the local micro-environment of active site can overcome the slow kinetics and unfavorable thermodynamics of CO$_2$ electroreduction. However, simultaneous optimization of both kinetics and thermodynamics is hindered by synthetic constraints and poor mechanistic understanding. Here we leverage laser-assisted manufacturing for synthesizing Cu$_x$O bipyramids with controlled tip angles and abundant nanograins, and elucidate the mechanism of the relationship between electron transport/ion concentrations and electrocatalytic performance. Potassium/OH$^-$ adsorption tests and finite element simulations corroborate the contributions from strong electric field at the sharp tip. In situ Fourier transform infrared spectrometry and differential electrochemical mass spectrometry unveil the dynamic evolution of critical *CO/*OCCOH intermediates and product profiles, complemented with theoretical calculations that elucidate the thermodynamic contributions from improved coupling at the Cu$^+$/Cu$^{2+}$ interfaces. Through modulating the electron transport and ion concentrations, we achieve high Faradaic efficiency of 81% at ~900 mA cm$^{-2}$ for C$_{2+}$ products via CO$_2$RR. Similar enhancement is also observed for nitrate reduction reaction (NITRR), achieving 81.83 mg h$^{-1}$ ammonia yield rate per milligram catalyst. Coupling the CO$_2$RR and NITRR systems demonstrates the potential for valorizing flue gases and nitrate wastes, which suggests a practical approach for carbon-nitrogen cycling.

The escalating carbon and nitrogen emission from continuous consumption of fossil fuels and human activities have raised global concerns on energy and environmental crisis[1–3]. Developing sustainable routes to close the carbon and nitrogen cycles is critical in overcoming the aforementioned issues[4,5]. Electrocatalysis driven by renewable sources has emerged as a promising approach to mitigating these problems while simultaneously producing valuable chemicals[6–8]. For

example, electrochemical carbon dioxide (CO$_2$) reduction (CO$_2$RR) into hydrocarbons and oxygenates such as ethylene (C$_2$H$_4$) and ethanol (EtOH) are attractive due to their high energy densities and values in the chemical industry[9–15]. Similarly, electrocatalytic nitrate reduction reaction (NITRR) for the production of ammonia, a vital fertilizer and commodity chemical, represents a valuable and versatile electrocatalytic process for wastewater remediation[16–20]. These

A full list of affiliations appears at the end of the paper. ✉e-mail: minghuizhu@ecust.edu.cn; xi_shibo@ices.astar.edu.sg; biy@rice.edu; tangbenz@cuhk.edu.cn; ruquanye@cityu.edu.hk

electrochemical processes involve complex multielectrons transfer reactions, demanding advancement in both kinetics and thermodynamics. In the case of $CO_2RR$, the selectivity and reaction rate for $C_2$ products are still suboptimal for practical implementation, primarily due to the sluggish thermodynamics and kinetics of C–C coupling[21]. The electrocatalytic nitrogen conversion from nitrate to ammonia commonly involves an intricate process of nine protons and eight electrons $(NO_3^- + 9H^+ + 8e^- \rightarrow NH_3 + 3H_2O)$. Improving the ammonia selectivity and current densities is essential for large-scale production[22]. Therefore, further exploiting the potential of $CO_2RR$ and NITRR hinges on the discoveries of efficient and cost-effective electrocatalysts.

The electron transport/ion concentration near active sites and enrichment of active sites are the most effective ways to improve the catalytic activity in terms of kinetics and thermodynamics[23,24]. The electric field can accelerate electron charge transport and ion concentrations near the active site, optimize adsorption/desorption of intermediates, and regulate the reaction microenvironment to boost the multielectron reduction reaction[25–27]. It is reported that the local electric field induced by Au nanoneedle structure can improve the local $CO_2$ concentration to boost electrocatalytic $CO_2$ reduction to CO[26]. Moreover, interface engineering is one of the effective ways to tune the thermodynamic barriers of reactions. Various structures, including grain boundaries, edges and steps, can modulate the coordination number (CN) of surface atoms, thus enhancing the adsorption capability[24,28,29]. $Cu^0/Cu^{1+}$ interface is also regards as the efficient interface for $CO_2RR$ to $C_{2+}$ due to the improvement of CO dimerization[28,30–32], and many theoretical studies prove that the subsurface O stabilized surface $Cu^{\delta+}$ species indeed accelerate the $C_{2+}$ products. Qiao et al. have proposed that the feasibility of $Cu^{2+}$ sites for promoting *CO hydrogenation to facilitate the generation of *CHO intermediates instead of *CO dimerization on the single $Cu^{2+}$ sites[33] and the combined computations studies also evidence that the oxidized Cu surface more significantly facilitates C–C coupling. So $Cu^+/Cu^{2+}$ interface with a higher oxidation state may be more helpful to promoting *CHO intermediates formation[34]. Based on the above discussion, the synergy of electric field effect and interface engineering can potentially boost the mass turnover frequency of multielectron electroreduction via "one plus one is greater than two" effect. In terms of thermodynamics, $Cu^+/Cu^{2+}$ interface could efficiently offer active sites to promote $CO_2RR$ to $C_{2+}$, while the electric field-induced higher electron transfer rate and ion concentration could accelerate its electrocatalytic process at the kinetic level.

Here we elucidate the mechanism of how the synergistic role of electric field and interfaces work in bifunctional electrochemical $CO_2RR$ and NITRR through controlling electron transport and ion concentrations. We achieve remarkable yields of $C_{2+}$ products under high current density and Faradaic efficiency (FE) by furnishing efficient material transport and active sites availability. In a typical demonstration, a laser-induced protocol is developed for the self-assembly of defective $Cu_xO$ nanoparticles into nano-bipyramids with controllable tip angles. Spherical aberration corrected-scanning transmission electron microscopy (SAC-STEM) unveils the double-tip structure at an atomic scale, consisting of fault steps, boundaries, and surface voids induced by $Cu^+/Cu^{2+}$ interface. Finite element method (FEM) simulations and adsorbed $K^+/OH^-$ concentration tests prove adjustable local electric field at the sharp tips. The exquisite integrated structure of the sharp bipyramids leads to an extremely high $C_2H_4$ yield rate of up to 1.55 mmol h$^{-1}$ mg$^{-1}$, an improved $C_{2+}$ FE exceeding 81% at a partial current density of 665.9 mA cm$^{-2}$. In the case of NITRR, we attain an ammonia-synthesizing rate of 81.83 mg h$^{-1}$ or 4.81 mmol h$^{-1}$ per milligram catalyst, accompanied by an $NH_3$ partial current density exceeding 600 mA cm$^{-2}$. To understand the interfaces effects, we employ in situ Fourier transform infrared (FTIR) spectrometry and differential electrochemical mass spectrometry (DEMS) and density functional theory (DFT). These complementary techniques corroborate the pivotal contribution of electric field and interfaces in simultaneously enhancing the FE and $NH_3$ yield rate for multielectron reduction. It is the first reported bifunctional catalyst to couple $CO_2RR$ and NTIRR reduction systems, to achieve turning waste into treasure, which is an important step towards future industrial green chemistry.

## Result

### Synthesis and structure characterization of L-$Cu_xO$ catalysts

The copper oxide-based nano-bipyramid electrocatalyst with high tip curvature (L-$Cu_xO$-HC) was synthesized by pulsed laser ablation in liquid (PLAL). Supplementary Fig. 1. illustrates the formation of nano-bipyramids: a copper target immersed in water is transiently heated by a pulsed laser and transformed into a vapor and/or plasma state, which is then quenched by the cool water to a solid state. The fast heating/quenching kinetics cause abundant defects of stacking faults. To understand the growth process of L-$Cu_xO$-HC, we tracked its structural evolution during the synthetic process from ex situ scanning electron microscopy and transmission electron microscopy (SEM and TEM; Supplementary Figs. 2 and 3) images. At first, the PLAL procedure yielded dispersed nanoparticles characterized as amorphous copper oxide ($Cu_xO$) according to X-ray diffraction (XRD; Supplementary Fig. 2). The formation of oxide nanoparticles could be ascribed to the reaction of laser-stimulated Cu nanoparticles with dissolved oxygen in water to form $Cu_xO$ (10 min). Next, the $Cu_xO$ nanoparticles self-assembled into loosely interconnected agglomerates and further merged to form a double vertebral body structure (30 min), which kept growing and ripening to develop into $Cu_xO$ bundles eventually (60 min). Supplementary Movie 1 illustrates the formation of oxide nanoparticles by laser irradiation in water.

Transmission electron microscopy (TEM), high-angle annular dark-field scanning transmission electron microscopy (HAADF-STEM) and energy-dispersive X-ray spectroscopy (EDX) elemental mapping analyses confirm the bipyramid structure of L-$Cu_xO$-HC (Fig. 1a–c and Supplementary Figs. 4–7)). The spherical aberration correction transmission electron microscopy (SAC-STEM; Fig. 1d, e) images show that the surface consists of areas with different contrasts; electron energy loss spectroscopy (EELS) spectra of O K-edge and Cu L-kedge (Supplementary Fig. 8) reveals that the dark and bright domains correspond to CuO and $Cu_2O$ phases, respectively, which agrees with selected area electron diffraction (SAED) and X-ray diffraction (XRD) pattern results (Fig. 1h and Supplementary Fig. 9). The stacking faults induce atomic steps on the surface of L-$Cu_xO$-HC, which could be seen at the edge (Fig. 1f) and from the surface intensity profile (Fig. 1d, e). The intensity fluctuation along the blue frame in Fig. 1d reveals the surface steps of L-$Cu_xO$-HC (Fig. 1g). For comparison, we also prepared a suite of L-$Cu_xO$ with medium and low tip curvatures (15° for L-$Cu_xO$-HC; 30° for L-$Cu_xO$-MC; 150° for L-$Cu_xO$-LC) (Supplementary Figs. 10–15). Both L-$Cu_xO$-MC and L-$Cu_xO$-LC possess a mixed $Cu_xO$ phase, as revealed by the XRD patterns with the simultaneous existence of CuO and $Cu_2O$ signals.

### Electrochemical $CO_2RR$ Performance

To validate the enhancement of $CO_2RR$ by the electric field and interface engineering, we also synthesized Cu tip with a smooth interface by an anodic method[35] and used commercial CuO and Cu for comparison of electrochemical performance (Supplementary Figs. 16 and 17). We evaluated the $CO_2RR$ performance of these catalysts in a flow cell in 1 M KOH electrolyte (versus the reversible hydrogen electrode (RHE) and no iR-compensation). We detected and analyzed the products under different potentials through the online gas chromatograph (GC) and $^1$H nuclear magnetic resonance (NMR) (Supplementary Figs. 18–22).

The built-in electric field effect can be indicated by comparing L-$Cu_xO$ with different tip curvatures. Figure 2a–c compare the $CO_2RR$

performance of different samples. For L-$Cu_xO$-HC (Fig. 2a), the partial current density of $C_{2+}$ production reached 665.9 mA cm$^{-2}$ with a corresponding $FE_{C2+}$ of 81 ± 5% at −2.8 V. With an increasing tip curvature, the $FE_{C2+}$ decreases to 73% and 57% for L-$Cu_xO$-MC and L-$Cu_xO$-LC, respectively (Fig. 2b). The field effect is more pronounced when comparing the $C_2H_4$ yield; 1.55 mmol$^{-1}$ h$^{-1}$ cm$^{-2}$ was obtained on the L-$Cu_xO$-HC, which is 2-fold of L-$Cu_xO$-MC and 3-fold of L-$Cu_xO$-LC. L-$Cu_xO$-HC always maintains the highest $C_{2+}$ current density over the entire voltage range compared with other samples (Fig. 2c), highlighting the electric field effect. In addition, the L-$Cu_xO$-HC also exhibits good durability and works steadily under a high current density of 600 mA cm$^{-2}$ for 12 h with an average $C_2H_4$ selectivity of 55% (Fig. 2d and Supplementary Figs. 23–29).

The nanograins interface effect can be inferred by comparing the L-$Cu_xO$ samples to other $Cu_xO$ prepared from conventional methods, including C-Cu, C-CuO, and Cu tip (Fig. 2a–c). Specifically, all the L-$Cu_xO$ with different curvatures have a much better $FE_{C2+}$ than the Cu tip. For C-Cu and C-CuO, both show similar curvatures to L-$Cu_xO$-LC,

but their $FE_{C2+}$ are significantly smaller (Fig. 2b, c). The above comparisons suggest the rich nanograins interface is favorable for $C_{2+}$ generation during $CO_2RR$.

## Built-in electric field mechanism investigations

We first investigate the distribution of locally enhanced electric fields on L-$Cu_xO$ with diverse curvatures of 15°, 30°, 60° and 90°$^{-1}$ by FEM-based theoretical simulations performed on COMSOL multiphysics (Fig. 3a and Supplementary Fig. 30). We found that with decreasing tip curvature from 90° to 15°, the tip-concentrated electron density shows a 6-fold enhancement, resulting in a significantly enhanced electric field at the 15° tip (Fig. 3b).

We then experimentally evaluate the effect of tip curvatures in shaping the local environments. We performed the K$^+$ absorbing test by measuring the concentration of adsorbed K$^+$ on the electrodes (Fig. 3c). The results show that the three L-$Cu_xO$s have a higher K$^+$ concentration than the quasi-planar C-Cu, and the adsorbed K$^+$ concentration can be further enhanced with a sharper tip (L-$Cu_xO$-HC).

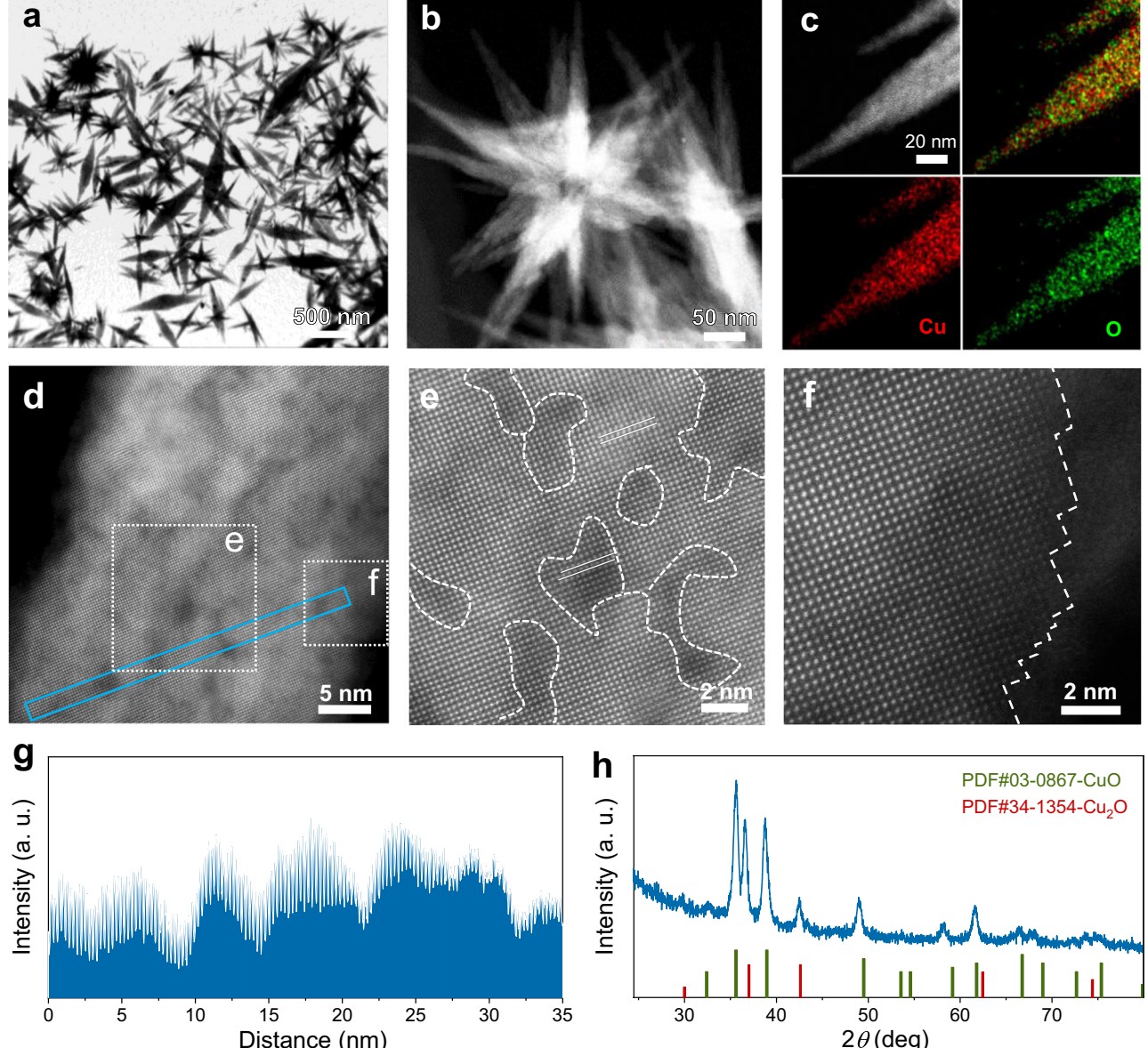

**Fig. 1 | Structural characterization of L-$Cu_xO$-HC. a** TEM, **b** STEM, and **c** corresponding EDX mapping images of L-$Cu_xO$-HC. **d** Spherical aberration-corrected high-resolution HAADF-STEM image. **e** and **f** are the enlarged area in **d**. **g** The line intensity profile acquired along the blue areas in **d**. **h** XRD pattern of L-$Cu_xO$-HC.

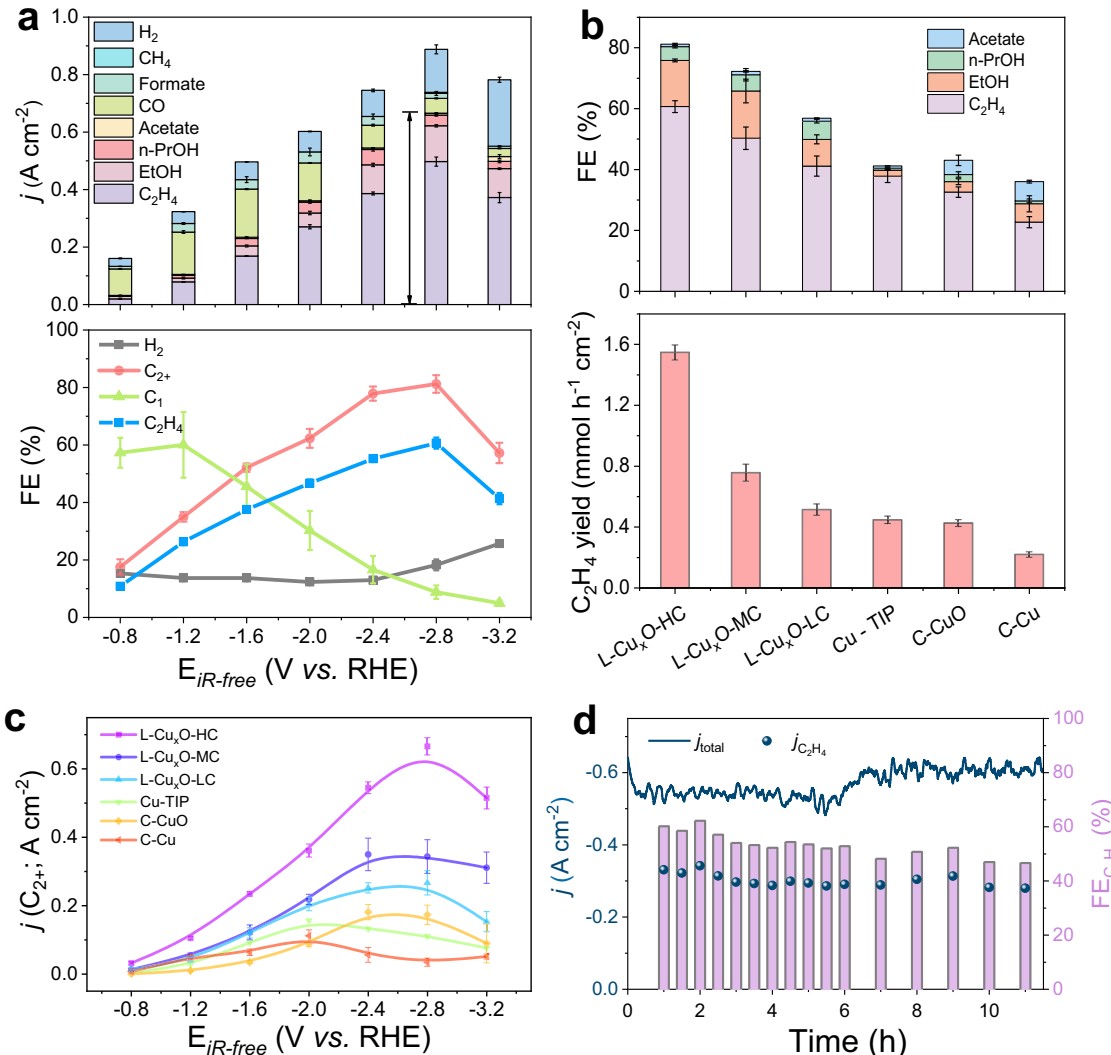

**Fig. 2 | CO₂ electroreduction performance. a** Current density (top) and FE values (bottom) of each CO₂RR product and H₂ on L-Cu$_x$O-HC at different applied potentials in 1 M KOH (without iR-correction). **b** FE values of C$_{2+}$ (top) and C$_2$H$_4$ yield production (bottom) for L-Cu$_x$O-HC, L-Cu$_x$O-MC, L-Cu$_x$O-LC, Cu-tip, C-CuO and C-Cu at their best performance. **c** C$_{2+}$ current density of each catalyst at different applied potentials. **d** Stability measurement in 1 M KOH under a high current density of -600 mA cm$^{-2}$. Error bars represent the standard deviation of three independent measurements.

This K$^+$ absorbing test result is consistent with the simulation results. It has also been reported that OH$^-$ species located at the catalyst surface could benefit C–C coupling and C$_{2+}$ production[36,37]. We thus use cyclic voltammetry (CV) to verify the OH$^-$ adsorption (OH$_{ad}$) features on L-Cu$_x$O. Figure 3d demonstrates the pronounced OH$_{ad}$ peaks associated with Cu (100) facets on L-Cu$_x$O-HC. Noted that there is a peak shifting with tip angles. The OH$_{ad}$ peaks shift to more negative potentials with decreasing tip curvatures, which suggests OH$^-$ could be adsorbed more easily on L-Cu$_x$O-HC, leading to a beneficial micro-environment for CO₂-to-C$_{2+}$ conversion (Supplementary Note 1). These results indicate that the local environments arising from the tip structure would concentrate the K$^+$ cation and OH$^-$ species near the catalyst surface, both of which could promote the C$_{2+}$ products.

Based on the above simulation and experimental results, we further broadened the CO₂RR performance to pH-universal conditions for L-Cu$_x$O-HC. Figure 3e, f and Supplementary Figs. 31 and 32 demonstrated the FE and C$_{2+}$ partial current density under acidic, neutral, and alkaline electrolytes. Notably, L-Cu$_x$O-HC achieves 72.8 ± 4.3% FE$_{C_{2+}}$ in neutral electrolytes and maintains 56.9 ± 5.4% FE$_{C_{2+}}$ in acidic electrolytes. The partial current density of C$_{2+}$ reaches

297.7 ± 24 mA cm$^{-2}$ and 397.1 ± 19 mA cm$^{-2}$ under acidic and neutral environments, respectively. Those good performances may be due to the good buffer capacity induced by the electric field effect of tip. The enhanced K$^+$ adsorption capacity will kinetically reduce the proton coverage on the Helmholtz plane through the competitive adsorption behavior driven by the electrostatic field[13], thus inhibiting the hydrogen evolution and contributing to the good performance in the pH-universal conditions (Fig. 3g, Supplementary Fig. 33 and Supplementary Table 1).

**Nanograins Cu$^+$/Cu$^{2+}$ interfaces mechanism investigations**

In addition to the electric field effect, the unique heterogeneous structures of L-Cu$_x$Os also benefit the C$_{2+}$ formation. To gain more accurate structural information about the chemical status and elemental composition of the samples, X-ray absorption spectroscopy at the Cu k-edge was used. As the X-ray absorption near-edge structure (XANES) result shows (Fig. 4a), the absorption edges of L-Cu$_x$O with different tip angles exhibit nearly identical profiles. These absorption edges lie between the spectrum features of C-CuO and C-Cu$_2$O, indicating the existence of the complex state of Cu$^+$ and Cu$^{2+}$ in the three L-Cu$_x$O electrocatalysts.

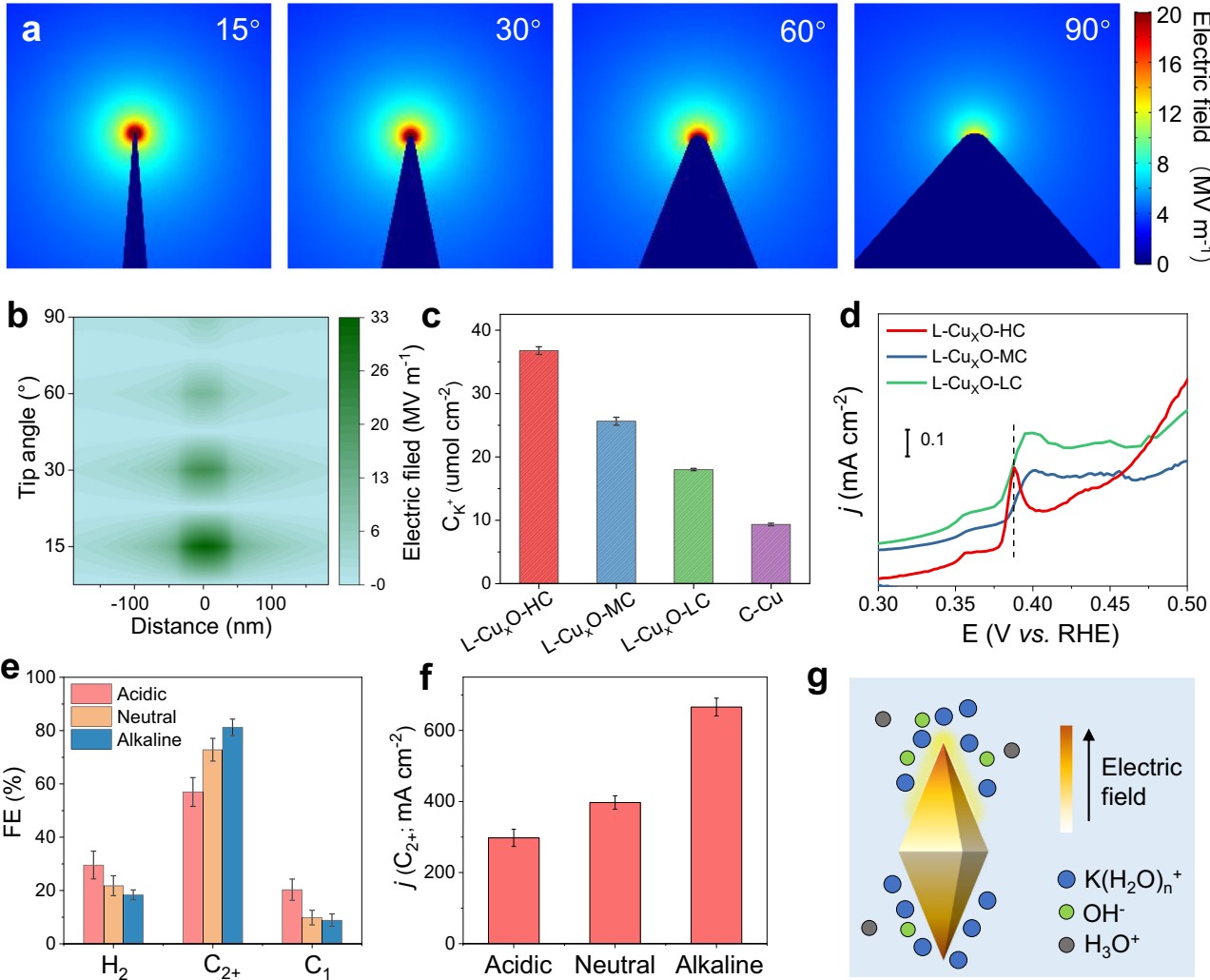

**Fig. 3 | Electric field detection and enhancement mechanism investigation.**
**a** Electric field distribution near the surface of 15°, 30°, 60° and 90° through
COMSOL multiple physical quantities. **b** The electric field values at the tips of
nanoneedle with different tip angle. **c** K⁺ absorption test. **d** OH⁻ absorption test. **e** FE
values and **f** $C_{2+}$ current density in acidic, neutral and alkaline environments.
**g** Schematic diagram. Error bars represent the standard deviation of three inde-
pendent measurements.

The extended X-ray absorption fine structure (EXAFS) reveals
the coordination numbers are about 3.6 (L-Cu$_x$O-HC), 3.4 (L-Cu$_x$O-
MC) and 2.8 (L-Cu$_x$O-LC) (Fig. 4b and Supplementary Table 2). This
might result from the increased Cu$_2$O phases that reduce the average
Cu-O coordination numbers. Wavelet transform was further applied
to investigate the coordination environment of the Cu species in
samples. As shown in Fig. 4c, the intensity of L-Cu$_x$O-HC, L-Cu$_x$O-MC,
and L-Cu$_x$O-LC are very close, confirming that the Cu-O bonds are
dominant in all these samples, matching well with the EXAFS fitting
results (Supplementary Fig. 34). These results agree with XANES and
XRD data that the Cu has the complex oxidation among these
catalysts.

To probe the dynamic evolution of surface adsorptions during
CO$_2$RR and to elucidate the mechanism of boosted $C_{2+}$ selectivity, we
then performed in situ attenuated total reflection Fourier transform
infrared spectroscopy (ATR-FTIR) using L-Cu$_x$O-HC and C-CuO. For
ATR-FTIR, the downward band in the resulting spectra indicated the
formation of intermediates during CO$_2$RR, while the upward band
referred to the consumption/desorption of surface species. Starting
from −0.4 V, the ATR-FTIR spectra for the L-Cu$_x$O-HC catalyst exhibit
several new peaks compared to C-CuO (Fig. 4d). Specially, two peaks at
~1034 and ~1231 cm⁻¹ associated with *COH and *CHO, respectively,

are observed for L-Cu$_x$O-HC, which are the important intermediates
for $C_{2+}$ products, especially $C_2H_4$[38]. L-Cu$_x$O-HC also exhibits additional
peaks at ~1182, ~1126, and ~1301 cm⁻¹ attributed to the absorbed
*OCCOH and *OC$_2$H$_5$, which are different from those of C-CuO[39]. This
indicates that more key intermediates of *OCCOH and *OC$_2$H$_5$, favor-
ing the production of $C_{2+}$, are formed on L-Cu$_x$O-HC than on C-CuO
catalysts. The ATR-FTIR results have provided experimental evidence
for the defect and tip-assisted C−C coupling mechanism over the
L-Cu$_x$O-HC catalyst.

To characterize the dynamic product profiles during the CO$_2$RR, a
differential electrochemical mass spectrometry (DEMS) was con-
ducted (Supplementary Fig. 35). We performed a continuous 4-cycle
DEMS measurement while scanning the potential between 0 V and
−2.1 V; each cycle took about 420 s. Figure 4e compares the mass signal
intensity of $C_2H_4$, CO and CH$_4$ at m/z = 26, 28, and 15, respectively, as a
function of cycle number and time. We choose m/z = 26 for $C_2H_4$ to
avoid interference from CO. With the increasing overpotential, CO is
produced first and then the CO production rate decreases when the
$C_2H_4$ emerges; the rate is further reduced with the appearance of CH$_4$,
which strongly proves CO is an essential intermediate in the produc-
tion of $C_2H_4$ and CH$_4$. In addition, the ratio of integrated mass signal
intensity of CO relative to the sum area of CO and $C_2H_4$ production on

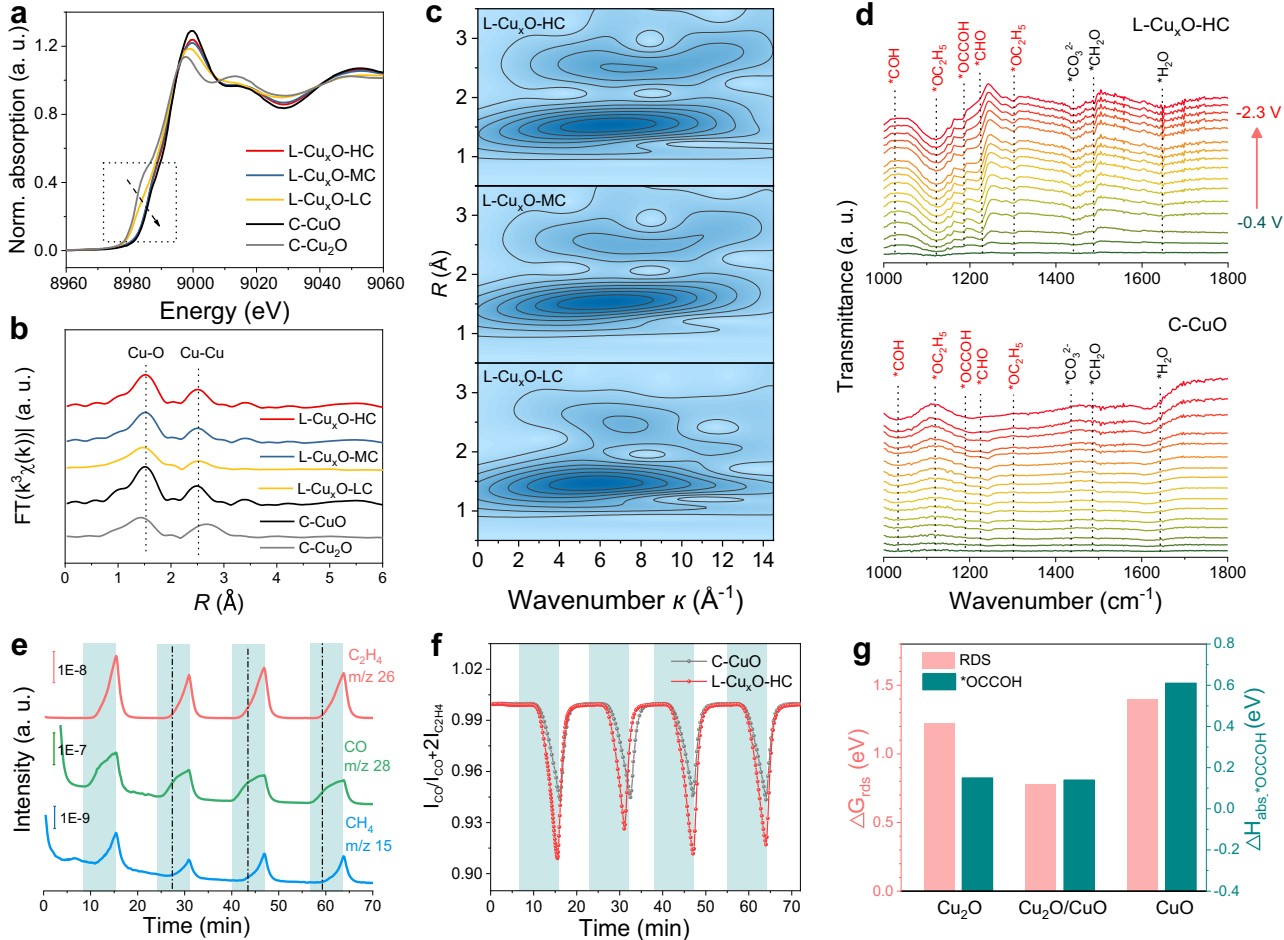

**Fig. 4 | In situ ATR-FTIR and DEMS measurement and mechanism. a** Normalized XANES, **b** Fourier transformed EXAFS and **c** WT−EXAFS Cu K-edge spectra of L-Cu$_x$O-HC, L-Cu$_x$O-MC, L-Cu$_x$O-LC, C-Cu$_2$O and C-Cu. **d** In situ ATR-FTIR obtained during chronopotentiometry in a potential window from −0.4 V to −2.3 V for L-Cu$_x$O-HC (top) and C-CuO (bottom) under CO$_2$RR. **e** Online DEMS over L-Cu$_x$O-HC and **f** I$_{NO}$/I$_{NH2OH}$ + I$_{NO}$ ratio at L-Cu$_x$O-HC and C-CuO (The shaded area represents the voltage application interval.). **g** DFT simulation and the free energy of the rate-determining step for C$_2$H$_4$ pathway and the adsorption energy of the *OCCOH intermediate on the Cu$_2$O/CuO interface, CuO and Cu$_2$O.

L-Cu$_x$O-HC dropped down sharply with the increasing overpotential, which indicates that CO is consumed more rapidly on L-Cu$_x$O-HC than C-CuO during CO$_2$RR, supporting that the tip effect is more significant than the C-CuO (Fig. 4f).

To further verify the role of the nanograins interfaces for C−C coupling over L-Cu$_x$O-HC, we used DFT to calculate the energy barriers for the C$_2$H$_4$ formation pathway. Here, Cu$_2$O(110) and CuO(110) slabs are first constructed as the nanograins of L-Cu$_x$O-HC based on the STEM and XRD results (Fig. 4g and Supplementary Figs. 36−40). Then, the energy barrier of each reaction step on the Cu$_2$O(110) slab, CuO(110) slab and Cu$_2$O(110)/CuO(110) interfaces are calculated to evaluate the catalytic performance of different catalytic sites in L-Cu$_x$O-HC. The full reaction pathway (Supplementary Fig. 36) shows that the rate-determining step (RDS) of Cu$_2$O(110)/CuO(110), Cu$_2$O(110) and CuO(110) are the same, which is *CO + *CO → *CO + *COH, but it is smallest on Cu$_2$O(110)/CuO(110) (0.78 eV), comparing with Cu$_2$O(110) (1.22 eV) and CuO(110) (1.4 eV). (Fig. 4g). Hence, C$_2$H$_4$ production happens more easily on the Cu$_2$O(110)/CuO(110) interface, indicating that these interfaces are more catalytically active in L-CuxO-HC. The *CO−*COH dimerization pathway has been reported to be highly essential over other pathways, particularly for C$_2$ products involving C−C coupling[40−43]. Furthermore, DFT calculations reveal that the adsorption energy of the *OCCOH intermediate is 0.14 eV on Cu$_2$O(110)/CuO(110), which is smaller than that on

Cu$_2$O(110) (0.15 eV) and CuO(110) (0.61 eV) (Fig. 4g). We further consider the pH effects on the reactions, which show similar trends with a lower energy barrier at the CuO/Cu$_2$O interfaces (Supplementary Fig. 37). These results verify that the presence of abundant interfaces is beneficial to the adsorption of the post-dimerization intermediate (*OCCOH), thus reducing the energy barrier of C−C dimerization. Our calculation mutually agreed with the in situ FTIR and online DEMS analysis.

### Electrochemical NITRR reduction performance and mechanism investigations

The field effect and nanograins interfaces might also benefit other multielectron electrochemical reactions. As a proof-of-concept application, the electrocatalytic NITRR activity of catalysts was evaluated using L-Cu$_x$O-HC in an H-type cell (Supplementary Figs. 41−48). Figure 5a shows the LSV curves of L-Cu$_x$O-HC and glass carbon in 1 M KOH with and without 1 M KNO$_3$. Glass carbon electrode shows similar currents in solutions with and without 1 M KNO$_3$, mainly attributed to hydrogen evolution reaction (HER). In 1 M KNO$_3$ solution, the current density of the glass carbon is slightly higher in KNO$_3$-containing solution because of carbon defects, but the total current remains negligible[22,44]. Notably, when using the L-Cu$_x$O-HC catalyst, the onset potential is significantly reduced to +0.2 V and the current density reaches >1000 mA cm$^{-2}$ at −1.2 V. Chrono-

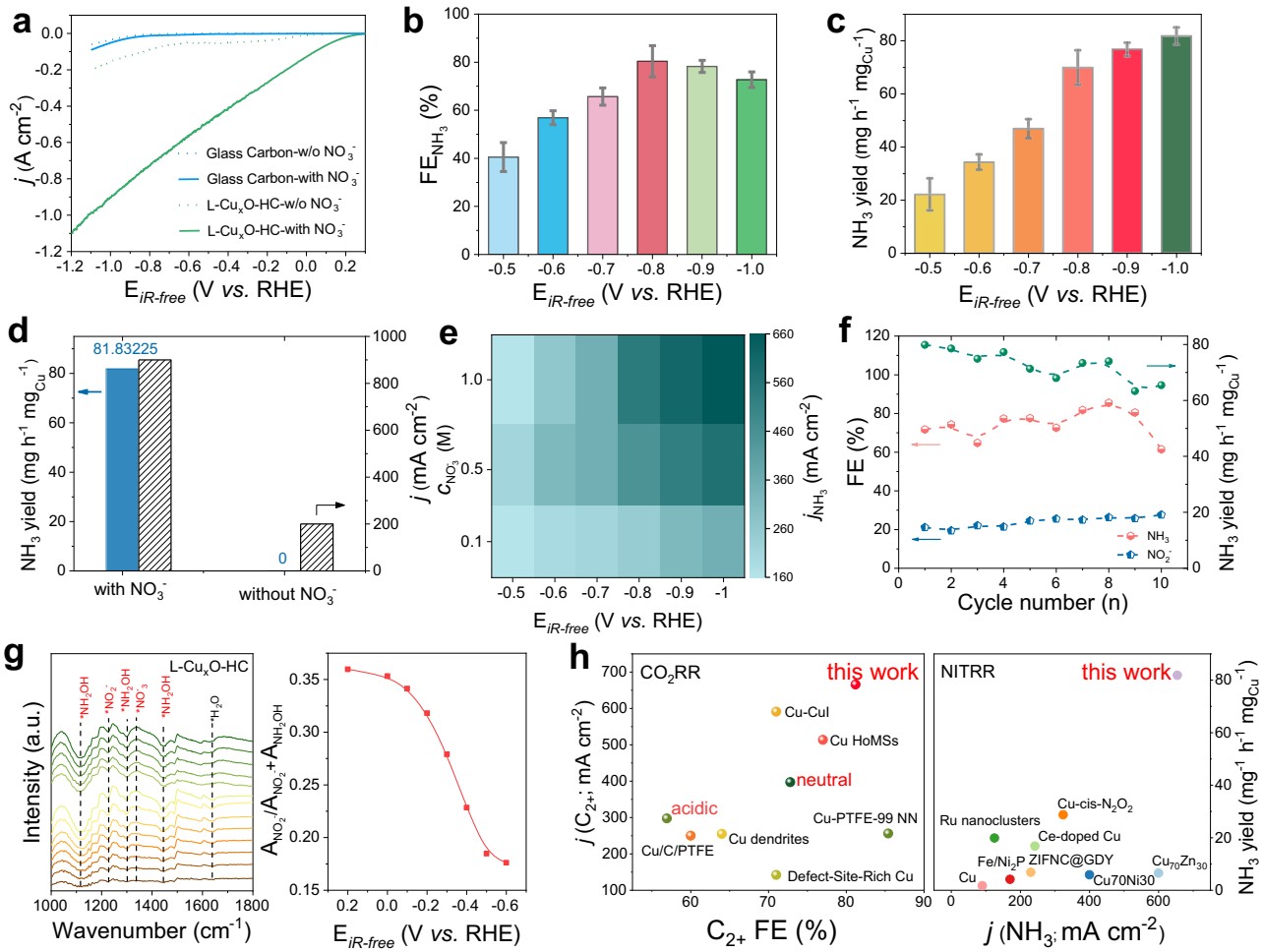

**Fig. 5 | Electrocatalytic NITRR performance and mechanism. a** Linear scan voltammetry curves of L-Cu$_x$O-HC and glass carbon in 1 M KOH and 1 M KOH + 1 M KNO$_3$. **b** FE of NH$_3$ at different potentials. **c** NH$_3$ yield rate of L-Cu$_x$O-HC at different potentials. **d** NH$_3$ yield rate and current density of L-Cu$_x$O-HC in 1 M KOH + 1 M KNO$_3$ and 1 M KOH. **e** Partial NH$_3$ current densities of L-Cu$_x$O-HC in 1 M KOH + 0.1 M KNO$_3$, 1 M KOH + 0.5 M KNO$_3$ and 1 M KOH + 1 M KNO$_3$. **f** Stability test on L-Cu$_x$O-HC at −0.8 V vs. RHE for 10 cycles. **g** In situ ATR-FTIR measurement on L-Cu$_x$O-HC during NITRR and A$_{NO2-}$/A$_{NO2-}$ + A$_{NH2OH}$. **h** Comparison of FE, product partial current density and yield with other reported catalysts (see the Supplementary Information for detailed references). Error bars represent the standard deviation of three independent measurements.

amperometry between −0.5 V and −1.0 V is performed; the NH$_3$ detected by $^1$H nuclear magnetic resonance (NMR) and the FE are summarized in Fig. 5b. Meanwhile, no N$_2$, NO, N$_2$O and NO$_2$ can be detected at all considered potentials according to the online electrochemical mass spectrometry (OEMS) result for the possible gaseous products (Supplementary Fig. 43). L-Cu$_x$O-HC reaches >80% FE$_{NH3}$ at the window from −0.8 V to −1.0 V and an NH$_3$ yield rate of 81.8 mg$^{-1}$ h$^{-1}$ mg$_{Cu}^{-1}$ at −1.0 V (Fig. 5c). We examine NH$_3$ in nitrate-free electrolyte and confirm that all our NH$_3$ products are generated by NITRR rather than by any contaminations (Fig. 5d). We also test isotope-labeled before and after NITRR coupled to future ensures the source of NH$_3$ (Supplementary Fig. 49). As different wastewater may have a varied nitrate concentration, we further examine the L-Cu$_x$O-HC catalytic performance with different KNO$_3$ concentrations (Fig. 5e and Supplementary Figs. 50 and 51). L-Cu$_x$O-HC preserves its high ammonia FE and activity under 0.1 M and 0.5 M KNO$_3$ and delivers an NH$_3$ partial current density of 353 and 573 mA cm$^{-2}$ at −1.0 V, respectively. The FE(NH$_3$) decreases with increasing KNO$_3$ concentrations from 0.1 M to 1 M. Notably, the FE$_{NH3}$ could reach ~95% from −0.5 V to −1.0 V in 0.1 M KNO$_3$. The higher selectivity in lower concentrations could be attributed to the competing adsorption between NO$_3^-$ and NO$_2^-$. During NITRR, NO$_3^-$ is first reduced to NO$_2^-$ and then undergoes deoxygenation and hydrogenation to form

NH$_3$. When NO$_3^-$ is in higher concentration, the first step will compete with the subsequential reduction, leading to a higher FE$_{NO2-}$ in 1 M KNO$_3$ (Supplementary Fig. 52). To probe the durability, a stability test was performed at −1.0 V for 10 consecutive electrolysis cycles; each with refreshed electrolyte and lasting 30 min. The current density keeps relatively steady at ~750 mA cm$^{-2}$ with a slight decrease (Supplementary Fig. 53). The FE$_{NH3}$ and yield rate show negligible decay over the whole test, implying high stability of the catalyst (Fig. 5f).

We further used in situ FTIR to track intermediates adsorbed on the surface. In Fig. 5g, five obvious absorption bands appear in the spectra of L-Cu$_x$O-HC[20,44]. Firstly, with the potential increased, the upward absorption bands at 1354 cm$^{-1}$ ascribe respectively to symmetric and asymmetric N-O stretching of NO$_3^-$, indicating consumption of NO$_3^-$; at the same time, the downward band at 1236 cm$^{-1}$ is attributed to N-O antisymmetric stretching vibration of NO$_2^-$, indicating NO$_2^-$ formation from NO$_3^-$ reduction; with potential negatively moving to 0.2 V, close to the onset potential of the LSV curve, another intermediate observed around 1110 cm$^{-1}$ was ascribed to N-O stretching vibration of hydroxylamine (NH$_2$OH), which is a key intermediate for NH$_3$ formation. In addition, with the negative shift of the working potential, the ratio of area of NO$_2^-$ relative to the sum area of NO$_2^-$ and NH$_2$OH production on L-Cu$_x$O-HC dropped sharply, indicating that the

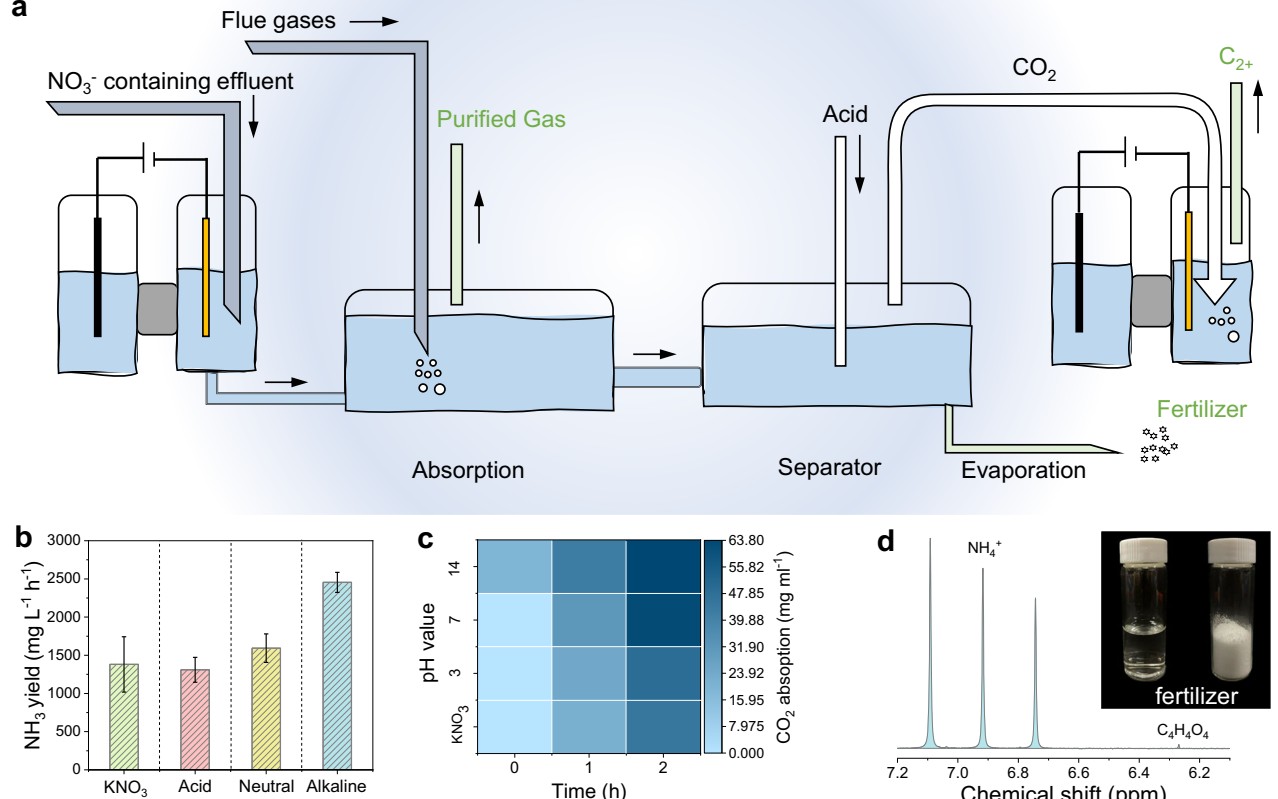

**Fig. 6 | Practical application of the NITRR and $CO_2$RR combined system.**
**a** Schematic diagram of electrochemical NITRR and $CO_2$RR system. **b** $NH_3$ production rate in different pH electrolytes. **c** $CO_2$ capture capacity of different electrolyte configurations. **d** Final product of ammonia fertilizer from the combined system. Error bars represent the standard deviation of three independent measurements.

tip effect could deeply enhance the hydrogenation of *$NO_2$ to the final $NH_3$ product.

We compare the FE, product partial current density and yield rate with other reported catalysts shown in Fig. 5h. It shows that L-$Cu_x$O-HC exhibits excellent performance in both $CO_2$RR and NITRR, attributed to the unique morphology and abundant nanograins interfaces. To our knowledge, L-$Cu_x$O-HC is among the best-reported catalysts with bifunctional activities (Supplementary tables. 3 and 4).

### Practical applications of the NITRR and $CO_2$RR

The chemical industry plays a key role in sustaining the world economy and underpinning future technologies[2]. Green chemistry aims to design chemical products and processes that reduce or eliminate the use or generation of hazardous substances[3,5]. Flue gases from fossil fuel consumption and nitrate waste from industrial water are two pollutants that cause the imbalance of carbon and nitrogen cycles. We further show that L-$Cu_x$O-HC, as a bifunctional catalyst, could be used for the cascade valorization of nitrate waste and flue gas. Specifically, an alkaline ammonia solution from NITRR could be used to capture $CO_2$ from flue gases. The addition of acid will form ammonium fertilizer and potash fertilizer (Fig. 6a) while releasing $CO_2$ for further conversion into $C_{2+}$ chemicals. Thus, coupling the NITRR-$CO_2$RR system reduces the chemical discharge that could be attractive to the environment and economics for wastewater and waste gas treatment.

Since industrial wastewater and polluted groundwater are mostly complex and the solution has a wide range of acidity and alkalinity, we assumed four different scenarios, including pure nitrate, acid (pH 3), neutral (pH 7), and alkaline (pH 14). After electrolysis at ~600 mA $cm^{-2}$ for 1 h, the ammonia yield rate reaches 1380.97, 1309.59, 1593.17 and 2454.97 mg $L^{-1}$ $h^{-1}$, and their $CO_2$ capture values are ~22 mg $mL^{-1}$, ~24.2 and 30.8 mg $mL^{-1}$ and 41.8 mg $mL^{-1}$, respectively. The $CO_2$ captured by the initial $NO_3^-$ electrolytes is negligible; only alkaline have the ability of $CO_2$ capture, which also proves that the treated electrolyte after NITRR is the main absorbent of $CO_2$ (Fig. 6b, c). Finally, the ammonia fertilizer and potash fertilizer are collected by rotary evaporation (Fig. 6d and Supplementary Figs. 54–56).

### Discussion

In conclusion, our study presents a laser-based method to design bipyramid catalysts with tunable tip curvatures and abundant nanograins interfaces, resulting in improved kinetics and thermodynamics of multielectron reduction. The sharp tip geometry induces a localized strong electric field, improving electron transport and ion concentration to regulate the reaction microenvironment in the kinetic way. Simultaneously, the abundant $Cu^+/Cu^{2+}$ interfaces provide plenty of active sites and lower the reaction barrier to contribute to $C_{2+}$ pathways by boosting the formation of *OC-COH, thus improving the reduction in the thermodynamics way. The combination of two effects leads to excellent catalytic performance. Notably, we achieve a prominent $C_{2+}$ partial current density of 665.9 mA $cm^{-2}$ with an FE of 81% for $CO_2$RR, and an $NH_3$ yield rate of 81.83 mg $h^{-1}$ $mg^{-1}$ with a partial current density exceeding 600 mA $cm^{-2}$ for NITRR. These values are among the highest reported values for $C_{2+}$ and $NH_3$ production. We propose that L-$Cu_x$O-HC, as a bifunctional catalyst, holds potential for application in the green chemical industry, enabling the transformation of wastewater and waste gas into valuable products. Our comprehensive in situ characterizations, combined with finite element simulation and density functional theory calculations, provide insights into the locally enhanced electric field and interfaces active sites, which are crucial for simultaneously improving both the $C_{2+}$ FE and yield rate of

$CO_2RR$. Similarly, these unique structural features also contribute to the outstanding $NH_3$ production from NITRR. Our laser-based synthesis approach, combining the dual effects of kinetics and thermodynamics, will inspire the design of next-generation multi-electron reduction catalysts for carbon and nitrogen cycling.

# Methods

## Material synthesis

**Preparation of L-Cu$_x$O-HC.** The synthesis of L-Cu$_x$O-HC was adapted from a previous report[45]. A copper target immersed in deionized water was ablated using a pulsed Nd:YAG laser (Nimma-600, Beamtech). The laser parameters were set as follows: pulse width 7 fs, wavelength 1064 nm, and pulse frequency 15 Hz. The copper raw material with a purity of 99.99% is made into a square target with a length of 20 mm and a thickness of 5 mm. The target was placed in a 50 ml beaker containing deionized water which surface was 10 mm higher than the target surface. The laser ablation lasted for 60 min to obtain a colloid solution with Cu NPs, and then the products was enriched by a centrifuge at $13,320 \times g$ for 15 min three times and then dry to obtain the L-Cu$_x$O-HC powder.

**Preparation of L-Cu$_x$O-MC and L-Cu$_x$O-LC.** The L-Cu$_x$O-MC and L-Cu$_x$O-LC were prepared by following the same procedure as for the L-Cu$_x$O-HC NPs, except that the Cu target was placed in a 200 ml beaker, containing deionized water and then laser ablation lasted for 3 h and 1 h, respectively.

**Preparation of Cu-tip.** Cu-tip electrodes were prepared by an anodized method[35]. The copper foil treated with 85% $H_3PO_4$ solution was used as the working electrode, and the Ag/AgCl platinum plate was used as the reference and counter electrodes. The copper foil was electropolished by applying a constant voltage of 4 V for 30 min, and then rinsing the electrodes with deionized water. Polished copper foil was used as the working electrode, platinum plate as the counter electrode, Ag/AgCl as the reference electrode, and the copper foil was anodized in 3 M KOH electrolyte. Cu-tip were synthesized by applying a constant current of 2 mA cm$^{-2}$ for 30 min, and then ultrasonically collected Cu-tip in ethanol to prepare Cu-tip GDL.

## Characterizations

The high-angle annular dark-field scanning transmission electron microscopy (HAADF-STEM) combined with Energy-dispersive X-ray spectroscopy (EDX) was employed to record the STEM and mapping images at an accelerating voltage of 200 kV. The X-ray diffraction (XRD) patterns of the samples were measured by a Bruker D2 using a Cu Kα source, with a scan step of 10° min$^{-1}$ and a scan range between 10° and 80°. The X-ray photoelectron spectroscopy (XPS) data were collected on a Thermo ESCALAB 250Xi spectrometer equipped with a monochromatic AlK radiation source (1486.6 eV, pass energy 20.0 eV). The data were calibrated with C 1s 284.8 eV. X-ray absorption fine structure (XAFS) measurements were performed in the transmission mode at beamline X-ray absorption fine structure for catalysis of the Singapore Synchrotron Light Source operated at 700 MeV with a beam current of 200 mA. The data processing was performed with the Athena and Artemis software packages. The in situ electrochemical FTIR measurements were performed by using a Thermo iS50. The online DEMS was provided by online analysis of produced intermediates and products of catalysts during $CO_2RR$ in flow cell. The concentrations of adsorbed K$^+$ on electrodes were detected by using a Thermo Scientific ICS-600 ion chromatograph system.

## Electrochemical measurements

**Electrochemical reduction of carbon dioxide in a flow cell.** $CO_2$ reduction was conducted in a three-chamber flow cell. The $CO_2$ gas was supplied directly to the catalyst layer (cathode, working electrode). The $CO_2$ gas flow rate was controlled using a mass flow controller and set to 10 sccm. 1 M KOH solution was used as both the catholyte and the anolyte. A platinum plate was used as the anode (counter electrode). Peristaltic pumps were used to control the flow rate of the electrolytes at -10 ml min$^{-1}$. An AEM (FAA-3-PK-75, Fumatech) was used to separate the cathode and anode chambers. The Ag/AgCl (1 M KOH, Gaossunion Co., Ltd., Tianjin) was used as the reference electrode. Electrolysis experiments were conducted using chronoamperometry by a CHI 650 electrochemical workstation (Chenhua, Shanghai). For each measurement, products were quantified after the amount of electron flowing through the cathode stable and at least three replicates were conducted to obtain an average value with the standard deviation. It should be noted that iR correction was not performed. For the neutral and acid, they were prepared by the same procedure as alkaline, except that the electrolyte changes to 0.5 M $K_2SO_4$ and 0.01 M $H_3PO_4$ + Saturated $K_2SO_4$ for the neutral and acid, respectively.

**Electrochemical reduction of potassium nitrate in a H-cell.** All NITRR experiments were performed in a two-chamber H cell separated by an ion exchange membrane (Nafion 117) using a three-electrode system connected to a CHI 650 electrochemical workstation (Chenhua, Shanghai)[18]. The obtained L-Cu$_x$O-HC loading on 0.5 cm$^2$ glass carbon was used as the working electrode, with Hg/HgO and platinum plate as the reference electrode and counter electrode, respectively[18]. The cathode electrolyte and anode electrolyte were 15 mL mixed KOH/KNO$_3$ solution (with different configurations). Line scan voltammetry (LSV) was performed at a scan rate of 10 mV s$^{-1}$. The potentiostatic tests were carried out at different potentials for 3600 s with a stirring rate of 800 rpm and the potential range for measuring the $NH_3$ FEs and yield rates was from −0.5 V to −1.0 V vs. RHE with intervals of −0.1 V. All potentials were recorded against the reversible hydrogen electrode (RHE) using $E_{RHE} = E_{Hg/HgO} + 0.0591 \times pH + 0.098$. Isotope labeling experiments were performed at −0.8 V vs. RHE, and $^{15}NO_3^-$ was used as the electrolyte N source. Unless otherwise specified, all measurements were carried out in an environmental chamber and environmental temperature without iR-compensation.

## Product detection

The gas and liquid products under different potentials during $CO_2RR$ through the online gas chromatograph (GC) and nuclear magnetic resonance (NMR).

**Ammonia detection.** $^1$H nuclear magnetic resonance ($^1$H NMR) was recorded on an AVANCE III HD 300 system for detection of ammonia. First, 2 M HCl was added to adjust the pH of the collected electrolyte to a weak acid. Then maleic acid ($C_4H_4O_4$, 50 ppm) was used as the external standard, and the peak area ratio of $NH_4^+$ and maleic acid was used to calibrate the $NH_4^+$ standard curve. Isotopic labeling experiments are also measured via the same process.

**Nitrite detection.** The nitrite concentration was measured by UV−vis spectrophotometry according to a previous report[19]. First, a mixture of deionized water (50 ml), N-(1-naphthyl) ethylenediamine dihydrochloride (0.2 g), p-aminobenzene sulfonamide (4 g) and phosphoric acid (10 ml, $\rho = 1.685$ g ml$^{-1}$) was used as the color reagent. Before detecting, the collected electrolyte sample was diluted to the detection range. Next, 40 μl of the color reagent mixed with the 2.0 ml sample solution thoroughly, and rested at ambient conditions for 20 min. The absorption intensity at a wavelength of 540 nm was measured by ultraviolet-visible spectrophotometry (UV-2600) using a series of pre-prepared potassium nitrite standard solutions and linear fitting to calibrate the concentration-absorbance curve.

The Faradaic efficiency (FE), yield, and selectivity were calculated according to the following equation:

$$FE_{gas} = \frac{Q_{gas}}{Q_{total}} \times 100\% = \frac{n_{gas} \times N \times F}{j \times t} \times 100\% = \frac{\frac{P}{RT} \times N \times F}{j} \times v \times 100\%$$

$$(1)$$

$$FE_{liquid} = \frac{Q_{liquid}}{Q_{total}} \times 100\% = \frac{n_{liquid} \times N \times F}{j \times t} \times 100\%$$

$$(2)$$

$$Yield_{NH3\,or\,C2H4} = \frac{n_{NH3\,or\,C2H4} \times M_{NH3\,or\,C2H4}}{m_{Cu} \times t}$$

$$(3)$$

$$Selectivity_{NO3-} = \frac{FE_{NH3\,or\,NO2-}}{FE_{NH3} + FE_{NO2-}}$$

$$(4)$$

$n$ is the amount of product (mol), $N$ is the number of electron transferred to form a molecule of product; $F = 95200\,C\,mol^{-1}$ is Faraday constant, $P$ is the atmosphere pressure (Pa), $T = 298\,K$ is the temperature (K) and $R$ is the molar gas constant $= 8.31\,J\,(mol\,K)^{-1}$, $v$ is gas flow rate, $j$ is the total current, $t$ is the electrolysis time (s), $M$ is the Molar mass of product (g mol$^{-1}$), $m$ is the quality of copper (mg).

**K⁺ adsorption measurement.** The concentrations of adsorbed K$^+$ on electrodes were performed in 0.1 M KHCO$_3$ solution by using a three-electrode system and an ion chromatograph[25]. All the electrodes were run in 0.1 M KHCO$_3$ solution with an applied voltage at −1.0 V vs RHE. When the running time reached 200 s, the electrode was directly raised above the electrolyte and then transferred with voltage and immersed in 10 mL of pure water. Next, the applied potential was removed, and shaking the electrode lasted for 1 min in pure water, to enable the adsorbed K$^+$ on the surface of catalysts to be completely released into the pure water. The concentration of K$^+$ in the water was checked using an IC after repeating the above process 10 times.

**OH⁻ electro adsorption measurements.** In-situ OH$_{ads}$ studies were conducted by flowing Ar in the H-cell. First, CO$_2$ electrolysis was conducted at a constant potential of −0.5 V versus the RHE for 30 min by switching the gas feed to CO$_2$. Immediately after electrolysis, the gas feed was switched to Ar, and then cyclic voltammetry (20 mV s$^{-1}$) was performed.

**In-situ FTIR spectroscopy.** In-situ FTIR spectra were acquired in a three-electrode cell with a Thermo Scientific Nicolet iS50 equipped. Ag/AgCl and Pt wire were used as the reference electrode and counter electrode, respectively. 0.1 M CO$_2$-saturated KHCO$_3$ was taken as the electrolyte. Each spectrum was recorded by 32 scans at an 8 cm$^{-1}$ spectral resolution.

**Online DEMS.** The flow cell was used during the DEMS measurements. Carbon paper coated with L-Cu$_x$O-HC electrocatalysts, Ag/AgCl, and platinum plate were used as the working electrode, the reference electrode and the counter electrode, respectively. LSV technology was employed from 0 to −2.4 V at a scan rate of 5 mV s$^{-1}$ until the baseline kept steady. Then, the corresponding mass signals appeared. After the electrochemical test was over and the mass signal returned to baseline, the next cycle was started using the same test conditions to avoid accidental errors during DEMS measurements. After four cycles, the experiment ended.

**Practical applications of the NITRR and CO₂RR.** The different electrolytes were obtained close to 600 mA cm$^{-2}$ under NTIRR after 1 h in relative electrolysis. Then the CO$_2$ was injected at a flow rate of 10 mL min$^{-1}$ for 15 min into obtained 10 mL of NITRR electrolytes or

other prepared solution. The capture of CO$_2$ was confirmed by titration using a calibrated hydrochloric acid solution (1 M). First, adding 50 uL bromocresol green-methyl red to the obtained colorless solution, hydrochloric acid solution was added until the color of the solution changed from green to pink. The molar mass of hydrochloric acid contained was equal to that of the absorbed CO$_2$[18]. The absorption capacity of some solutions was estimated by three independent tests. After adding acid to neutralize the electrolyte, pass through a rotary evaporator to obtain ammonium mixture crystals, and then place it in an oven overnight to obtain the final product.

**Theoretical simulations of finite-element method (FEM).** The 2D models were constructed to simplify the theoretical model, which represent the surface of the L-Cu$_x$O. The geometric dimension of the L-Cu$_x$O was obtained from the SEM. The electric field (E) distribution was described by the following equation,

$$E = -\nabla V$$

$$(5)$$

$$\rho = \varepsilon_r \varepsilon_0 \nabla \cdot E$$

$$(6)$$

where $V$, $\rho$, $\varepsilon_O$, and $\varepsilon_r$ represent the applied potential bias, charge density, dielectric in vacuum and materials, respectively. $E$ was the negative gradient of the electric potential.

The related materials' electrical conductivities were set according to the previous reference and the boundary condition was set as follows:

$$\vec{n} \cdot J = 0$$

$$(7)$$

The ion absorption behavior was described by the Poisson–Nernst–Planck equations,

$$\nabla \cdot \left( D\nabla c_i + \frac{Dz_i e}{k_B T} c_i \nabla V \right) = 0$$

$$(8)$$

where $c$, $D$, $z$, $e$, $k_B$, and $T$ are the ion concentrations, diffusion coefficients, ion valences, the elementary charge, Boltzmann constant, and the absolute temperature, respectively[46].

To ensure the high accuracy of the simulation, the densest conventional triangular meshes were used for all simulations, where the maximum element size was 1 nm. The maximum element size was set as 10 nm in the bulk electrolyte.

**Computational methods.** The density functional theory (DFT) calculations were conducted by the Vienna ab initio simulation package (VASP)[47]. Projector augmented wave (PAW) pseudopotential is applied for the core electrons and the generalized gradient approximation (GGA) in the form of Perdew−Burke−Ernzerhof (PBE) for the exchange-correlation potentials[48,49]. The cutoff energy is 450 eV for the valence electrons. The atoms were fully relaxed until the energy convergence reached 0.00001 eV, and the force acting on each atom was less than 0.02 eV/Å. The Van der Waals interaction was calculated at the DFT-D3 level as proposed by Grimme with the IVDW value of 12[50,51]. Initial structures of CuO and Cu$_2$O were obtained from the Materials Project database based on our XRD and TEM results[52]. After building the slab models, a vacuum layer of 15 Å was added to avoid interactions between adjacent images. These slab models' bottom two atomic layers were kept fixed during the simulations.

The CO$_2$ and NO$_3^-$ reduction reactions were modeled based on the computational hydrogen electrode model (CHE)[53], where the free energy of proton and electron pair is half of the free energy of free hydrogen gas molecule, which means, $G(H^+ + e^-) = 0.5\,G(H_2)$, when pH = 0. The Gibbs free energy change for each reaction step is

calculated as,

$$\Delta G = \sum G(products) - \sum G(reactants) \qquad (9)$$

Where $G(i)$ is the Gibbs free energy of species $i$. Gibbs free energy of each species was calculated as $G = E + ZPE - TS$, where $E$ is the total energy obtained from DFT calculations, $ZPE$ is the zero-point energy, and S is the entropy. Temperature $T$ was set to be 300 K. We further studied several key reaction steps considering the solvent effect and the pH effect. The solvent effect was studied with the code VASPsol using the Poisson–Boltzmann implicit solvation model with a dielectric constant $\varepsilon = 78.4$ for water. To include the pH effect, for reaction with proton involved, a correction term $\Delta G_{pH} = 0.059 \times pH$ was added to the right side of the above equation.

## Data availability

All data generated or analyzed during this study are included in this published article and Supplementary Information files) or can be obtained from the authors on reasonable request.

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

## Acknowledgements

R.Y. acknowledges support from Guangdong Basic and Applied Basic Research Fund (2022A1515011333), Hong Kong Research Grant Council (11309723), the Shenzhen Science and Technology Program (JCYJ20220818101204009) and State Key Laboratory of Marine Pollution (SKLMP/IRF/0029). B.Z.T. acknowledges support from Shenzhen Key Laboratory of Functional Aggregate Materials (ZDSYS20211021111400001), the Science Technology Innovation Commission of Shenzhen Municipality (KQTD20210811090142053, JCYJ20220818103007014).

## Author contributions

R.Y. conceived and designed the research. R.Y., B.T. and B.Y. supervised the research. W.G. carried out most of the experiments and S.Z. performed the calculations. H.W. and M.Z. performed the in situ Fourier transform infrared studies. S.X. performed the X-ray absorption spectroscopy experiment and analysis. J.Z., Y.M., Y.S., L.Cheng, L.Chang, G.L., Y.L., G.W., L.G. and X.W. conducted part of the experiments. R.Y. and W.G. analyzed the data and wrote the manuscript with input from the other authors.

## Competing interests

The authors declare no competing interests

## Additional information

[1]Department of Chemistry, State Key Laboratory of Marine Pollution, City University of Hong Kong, Hong Kong 999077, China. [2]City University of Hong Kong Shenzhen Research Institute, Shenzhen 518057 Guangdong, China. [3]Department of Chemistry and the Hong Kong Branch of Chinese National Engineering Research Center for Tissue Restoration and Reconstruction, The Hong Kong University of Science and Technology, Hong Kong 999077, China. [4]Department of Materials Science and Nano Engineering, Rice University, 6100 Main Street, Houston, TX 77005, USA. [5]State Key Laboratory of Chemical Engineering, East China University of Science and Technology, Shanghai 200237, China. [6]Institute of Materials Research, Tsinghua Shenzhen International Graduate School, Tsinghua University, Shenzhen 518055 Guangdong, China. [7]Institute of Chemical and Engineering Sciences, A*STAR, Singapore 627833, Singapore. [8]School of Energy and Environment, City University of Hong Kong, Hong Kong 999077, China. [9]School of Science and Engineering, Shenzhen Institute of Aggregate Science and Technology, The Chinese University of Hong Kong, Shenzhen 518172 Guangdong, China. [10]These authors contributed equally: Weihua Guo, Siwei Zhang. ✉e-mail: minghuizhu@ecust.edu.cn; xi_shibo@ices.astar.edu.sg; biy@rice.edu; tangbenz@cuhk.edu.cn; ruquanye@cityu.edu.hk

