## [Peer Review File · Nature Communications]

REVIEWER COMMENTS

Reviewer #1 (Remarks to the Author):

This manuscript reports the potential of using CuxO bipyramids with controlled tip angles and abundant nanograins towards the CO₂RR and NITRR. Detailed experimental analysis and theoretical studies are conducted to elucidate the origin for the high CO₂RR/NITRR reactivity of CuxO. The presented results represent some achievements in the field of CO₂RR/NITRR electrocatalysis in terms of new catalyst, fundamental understanding of the catalytic mechanism and impressive performances, which will be attractive to broad audience. Some specific comments are provided as follows

1. During the NO₃RR electrolysis, many side-products may be formed, such as NO₂⁻, N₂, NO, N₂O etc, which should be measured and analyzed.
2. Why potassium hydroxide (1 M KOH) was used as the electrolyte in this work? The gaseous NH₃ is known to be easily escaped from the solution under the alkaline condition.
3. The pH value of the electrolyte may be varied accompanied with ammonia production. How the catalytic stability can be maintained?
4. For DFT calculations, do the authors take solvent effect into account when calculating the free energy? Some packages allow for simulating the solvation effect and pH effect on DFT calculations; they should be considered to provide a deeper understanding of the reaction mechanism.
5. The applied potential (-0.8 V. RHE) for the optimum NH₃-FE is relatively high compared to other reported eNO_xRR electrocatalyst. The high applied potential is not beneficial in aspect of techno-economic.

Reviewer #2 (Remarks to the Author):

In the manuscript currently under review, Guo et al. present their findings on the utilization of CuxO bipyramids as catalysts for the electrochemical reduction of carbon dioxide (CO₂RR) and nitrogen (NITRR). The authors report a great achievement of an 81% FE for C₂⁺ products at 900 mA cm⁻² in CO₂RR, as well as an ammonia yield rate of 81.83 mg h⁻¹ in NITRR. While the paper is well-written, the concept of materials design explored in this study is general common, as the role of the copper oxidation state in CO₂RR has been extensively investigated, including the exploration of various mixed valence states (Velasco-Vélez et al., ACS Sustain. Chem. Eng., 2018, 7(1): 1485-1492; Scholten et al., ACS Catalysis, 2019, 9(6): 5496-5502; Chou et al., J. Am. Chem. Soc., 2020, 142(6): 2857-2867). Therefore, this manuscript is not suitable for publication in the Nature Communications at this stage. In addition,

there are several critical aspects that remain unclear in this work, which hinder a comprehensive understanding of the results and conclusions presented.

1: The quality of the SEM image in Fig.S2 is insufficient to discern the growth trajectory of the purported Cu_xO bipyramids, as asserted by the authors. It is suggested that TEM be employed to provide a more detailed view of the material's morphology at various time intervals.

2. The approach of distinguishing between CuO and Cu_2O based on the brightness and darkness of the TEM image lacks rigorousness. It is advisable to employ Energy-Loss Near-Edge Structure (ELNES) analysis as a more reliable method for determining the valence state of 3d metals. Additionally, conducting Selected Area Electron Diffraction (SAED) is essential to substantiate the authors' conclusions.

3. In the present manuscript, an array of characterization techniques, including TEM, SEM and XRD, were utilized to probe the structures and compositions of Cu_xO bipyramids. Notwithstanding, these characterizations were exclusively focused on the Cu_xO bipyramids prior to catalyzing CO_2 reduction, with post-reaction analyses notably absent. The author declared the angular configuration of the tips influenced the distribution of the electric field, but it has been previously documented that the morphology of certain Cu-based nanocatalysts can undergo substantial modifications under significant reduced current during CO_2RR . Consequently, comprehensive physicochemical characterizations subsequent to electrolysis are imperative to ascertain the stability of these nanostructures.

4. The authors concluded that the $\text{Cu}^+/\text{Cu}^{2+}$ interface could offer more efficiently active sites to promote CO_2RR to C_2^+ . Although the notion that the oxidized state of Cu may be reduced to CuO during CO_2RR remains a contentious issue, given the diverse testing conditions employed by different researchers. It is noteworthy that the authors conducted their tests under a reduction current of 900 mA cm^{-2} , which raises questions regarding the ability of Cu to retain its valence state under such strenuous reducing conditions. Therefore, the execution of in situ EXAFS is recommended to ascertain the actual catalytic sites.

5. It would be appreciative if the authors could elucidate the principles of testing in Fig. 3d, supplemented with pertinent literature citations for further context and validation.

6. Some basic electrochemical characterization needs to be provided, such as electrochemical impedance spectroscopy, electrochemical stability testing for at least 10 hours, partial current density of the product, etc.

Reviewer #3 (Remarks to the Author):

Herein, the authors employed a laser-assisted technique to synthesize Cu_xO bipyramids for electrochemical CO_2RR . The sharp tip angle of the nanostructure favors cation accumulation which stabilizes reaction intermediates via field effect and therefore assists the reaction kinetics. Also, the partially oxidized Cu interface further favors the formation of a critical intermediate for C-C coupling.

Overall, the study is extensive, the hypothesis is well supported by various spectroscopic analyses and theoretical simulations. Furthermore, in addition to the fundamental scientific exploration, practical industrial application has also been demonstrated. I feel that the manuscript can be accepted in Nature Communications after the following minor concerns be addressed:

1. In Supplementary Fig. 21, the authors presented XPS and PXRD analysis of the potential driven transformation of the catalyst during electrochemical CO₂RR. The data suggests that the surface is still a partially oxidized Cu_xO structure. However, does the metallic Cu phase appear during electrolysis? Enhancement of Cu-Cu scattering feature in EXAFS and shift of Cu-K edge in Xanes after electrolysis would be able to answer this question. Also, Is there any difference in impedance before and after long-term electrolysis?

2. It would be interesting to see the CO₂RR trend with another alkali metal cation such as Li⁺. The interfacial accumulation of larger cations (K⁺, Cs⁺) is relatively higher than the smaller ones. However, does the electrostatic field at the tip curvature assist the accumulation of smaller cations as well? From a fundamental point of view, it would be interesting to observe if the tip-assisted accumulation diminishes the cation size effect.

Response to Reviewers' Comments

Reviewer #1

Comment 1: This manuscript reports the potential of using Cu_xO bipyramids with controlled tip angles and abundant nanograins towards the CO_2RR and NITRR . Detailed experimental analysis and theoretical studies are conducted to elucidate the origin for the high $\text{CO}_2\text{RR}/\text{NITRR}$ reactivity of Cu_xO . The presented results represent some achievements in the field of $\text{CO}_2\text{RR}/\text{NITRR}$ electrocatalysis in terms of new catalyst, fundamental understanding of the catalytic mechanism and impressive performances, which will be attractive to broad audience. Some specific comments are provided as follows

Response: We highly appreciate the Reviewer for the positive comments on the novelty of our work. We have tried our best to revise our manuscript according to the invaluable comments from the Reviewer.

Comment 2: 1. During the NO_3RR electrolysis, many side-products may be formed, such as NO_2^- , N_2 , NO , N_2O etc, which should be measured and analyzed.

Response: We thank the Reviewer very much for this very important suggestion. Based on the reviewer's suggestion, we have tried to detect the potential gas products N_2 , NO , N_2O and NO_2 , which were analyzed during the reaction by online electrochemical mass spectrometry (OEMS). We use a liquid flow cell as the reaction vessel with Ar as the carrier gas. The curves of the intensity of NH_3 , N_2 , NO , N_2O and NO_2 vs. time are shown in **Supplementary Fig. 43**, illustrating no detection of N_2 , NO , N_2O and NO_2 .

NO_2^- has been quantitatively detected by the Nessler Reagent method (**Supplementary Fig. 44**). The FE of NH_3 and NO_2^- is shown in **Supplementary Fig. 51**.

We have added some descriptions on this point, which are highlighted in red color on Page 9 in our revised manuscript, as shown below.

“Meanwhile, no N_2 , NO , N_2O and NO_2 can be detected at all considered potentials according to the online electrochemical mass spectrometry (OEMS) result for the possible gaseous products (Supplementary Fig. 43).”

We have also added the **Supplementary Fig. 43** about the OEMS result for the possible gaseous products of NITRR, which are highlighted on Page 35 in our revised Supplementary Information, as shown below.

Supplementary Fig. 43. The OEMS result for the possible gaseous products of NITRR (NH_3 (g), N_2 , NO , N_2O , NO_2).

Supplementary Fig. 44. Calibration curves. UV-vis calibration curves of (a, b) ammonia (NH_3) and (c, d) nitrite (NO_2^-), respectively.

Supplementary Fig. 51. Electrochemical performances. FE of NH_3 and NO_2^- in (a) 1 M KOH + 1 M KNO_3 (b) 1 M KOH + 0.5 M KNO_3 and (c, d) 1 M KOH + 0.1 M KNO_3 .

Comment 3: 2. Why potassium hydroxide (1 M KOH) was used as the electrolyte in this work? The gaseous NH_3 is known to be easily escaped from the solution under the alkaline condition.

Response: We thank the Reviewer for this important question. We choose potassium hydroxide (1 M KOH) as the electrolyte in this work for these four reasons. (1) The alkaline electrolyte is beneficial to inhibit the possible hydrogen evolution reaction, which is the competing reaction of NITRR. (2) The gaseous NH_3 escapes from the solution under the alkaline condition flowing this reaction equation: $\text{NH}_4^+ + \text{OH}^- \rightleftharpoons \text{NH}_3(\text{g}) + \text{H}_2\text{O}$. However, the solubility of ammonia in water is that 1 volume of water dissolves 700 volumes of ammonia, 1 liter of water dissolves 495.67g of ammonia gas. Our best ammonia production rate is $1.215 \text{ g L}^{-1} \text{ h}^{-1}$, which is greatly lower than the solubility limitation of the ammonia in water. Therefore, although ammonia in alkaline solutions is easily stripped out, the amount of ammonia gas escaping from the water in our work can be negligible. (3) Compared to other electrolytes, alkaline electrolytes can decrease the overpotential of the oxygen evolution reaction at the anode, thereby lowering the full cell voltage. Because high current density leads to the increase of cell voltage, especially OER, while alkaline electrolyte can greatly reduce the overpotential of the OER, thereby reducing the cell voltage. (4) When we

conducted the nitrate electroreduction reaction, we noticed some of the reported works use the alkaline electrolyte in their measurement.¹⁻⁴ To reasonably compare our results with the literature, we also used alkaline electrolyte. Based on the above reasons, we choose potassium hydroxide (1 M KOH) as the electrolyte in this work.

Comment 4: 3. The pH value of the electrolyte may be varied accompanied with ammonia production. How the catalytic stability can be maintained?

Response: We thank the Reviewer very much for raising this important question. In the process of nitrate electroreduction reaction, we did observe the change in pH value due to the formation of ammonia and ions. We tested the pH of the electrolyte before and after 1 hour of electrolysis at -0.8 V vs. RHE and found that the pH value increased from 13.73 to 13.88, as shown in **Figure R1**. We also observed that when it was tested for more than 1 hour during NITRR, the current density increased gradually. This might be related to the change of the applied potential vs RHE due to pH change. Based on this situation, when conducting the stability test, we refreshed the electrolyte every half hour to ensure the stable pH value and ion concentration of the solution, as shown in the **Supplementary Fig. 53**.

Figure R1. The change of pH value before and after NITRR reaction for 1 hour.

Supplementary Fig. 53. Electrochemical performances. Chronoamperometry during the cycles test of L-Cu_xO-HC.

Comment 5: 4. For DFT calculations, do the authors take solvent effect into account when calculating the free energy? Some packages allow for simulating the solvation effect and pH effect on DFT calculations; they should be considered to provide a deeper understanding of the reaction mechanism.

Response: We thank the Reviewer a lot for this important suggestion because it has been shown that solvent effect could change the activity of catalysts.⁵ As for the pH effect, it can help us understand the behavior of a catalyst in different pH environments. It is also worth mentioning that the energy correction term for the solvent effect and the pH effect is different for different adsorbates. Hence, to address the reviewer's concerns, we revisited the key step: $*CO + *CO \rightarrow *CO + *COH \rightarrow *OCCOH$ steps with the consideration of the solvent and the pH effect on Cu₂O, CuO, and the Cu₂O/CuO model. The solvent effect is included using the VASPsol package and the pH effect is considered by the correction term of $\Delta G_{pH}=0.059 \times pH$. The results are summarized in the following figures. Without considering the solvent effect, the ΔG of the step $*CO + *CO \rightarrow *CO + *COH$ for the Cu₂O, CuO, and the Cu₂O/CuO model is 1.22, 1.40, and 0.78 eV, respectively. With the consideration of the solvent effect, the ΔG of the step $*CO + *CO \rightarrow *CO + *COH$ is reduced to 0.93, 1.16, and 0.60 eV in pH=0, respectively. Therefore, the solvent effect stabilizes the $*CO + *COH$ intermediate, and the performance of the interface model is further improved. The ΔG of the step $*CO + *COH \rightarrow *OCCOH$ is previously 0.15, 0.61, and 0.14 eV, which changes to -0.02, 0.72, and 0.26 eV. When the pH increases, the ΔG of the step $*CO + *COH \rightarrow *OCCOH$ is not affected, as no proton is involved in this reaction. The ΔG of the step $*CO + *CO \rightarrow *CO + *COH$ continuously increases with the pH value increasing from 0 to 14. However, considering

the pH effect does not change the performance trend of these catalysts. We thank the reviewer again for the suggestions.

Based on the reviewer's suggestion, we have added **Supplementary Fig. 37** highlighted on Page 9 of manuscript and on Page 32 in Supplementary Information.

Supplementary Fig. 37. Reaction Gibbs free energy profile for $*CO + *CO \rightarrow *CO + *COH \rightarrow *OCCOH$ steps on Cu_2O , CuO , and the Cu_2O/CuO model at various pH.

Comment 6: 5. The applied potential (-0.8 V. RHE) for the optimum NH_3^- FE is relatively high compared to other reported eNO_xRR electrocatalyst. The high applied potential is not beneficial in aspect of techno-economic.

Response: We thank the Reviewer a lot for this important question. Our electrochemical measurements are all conducted at the potential without iR compensation during nitrate electroreduction reaction. However, based on our best knowledge, most of the reported work added the iR correction when reporting their potentials. Based on the reviewer's suggestion, we acquired the LSV curves with 50% and 80% compensation, respectively (**Supplementary Fig. 42**). According to the result, its best potential for the optimum NH_3^- FE is -0.6 V and -0.3 V with 50% and 80% iR correction, respectively, which is among the best performance^{1,2,6}.

We have also added **Supplementary Fig. 42** on Page 34 in our revised Supplementary Information.

Supplementary Fig. 42. Linear scan voltammetry curves of L-Cu_xO-HC in 1 M KOH + 1 M KNO₃ at different potential with iR-free, iR-50% compensation and iR-80% compensation.

Reviewer #2

Comment 1: In the manuscript currently under review, Guo et al. present their findings on the utilization of Cu_xO bipyramids as catalysts for the electrochemical reduction of carbon dioxide (CO_2RR) and nitrogen (NITRR). The authors report a great achievement of an 81% FE for C_{2+} products at 900 mA cm^{-2} in CO_2RR , as well as an ammonia yield rate of 81.83 mg h^{-1} in NITRR . While the paper is well-written, the concept of materials design explored in this study is general common, as the role of the copper oxidation state in CO_2RR has been extensively investigated, including the exploration of various mixed valence states (Velasco-Vélez et al., *ACS Sustain. Chem. Eng.*, 2018, 7(1): 1485-1492; Scholten et al., *ACS Catalysis*, 2019, 9(6): 5496-5502; Chou et al., *J. Am. Chem. Soc.*, 2020, 142(6): 2857-2867). Therefore, this manuscript is not suitable for publication in the *Nature Communications* at this stage. In addition, there are several critical aspects that remain unclear in this work, which hinder a comprehensive understanding of the results and conclusions presented.

Response: We thank the Reviewer for this comment. Although the role of the copper oxidation state in CO_2RR has been investigated, our materials design focus on synthesizing copper bipyramids, featured with tailorable tip angles and abundant nanograins. By harnessing the dual effects of strong localized electric field and abundant grain interfaces to regulate electron transport and ion concentration, we have enhanced the current density and selectivity of multielectron reactions at the same time. We demonstrate the advantages of this unique structure through CO_2 reduction and nitrate reduction reaction (CO_2RR and NITRR), achieving exceptional selectivity and current density simultaneously, even in acidic electrolytes, surpassing the limitations of previous approaches. To be specific, firstly, unlike conventional tips prepared through anodic oxidation with smooth surfaces, our innovative laser-based method produces copper-based bipyramids with abundant nanograins. Most of the reported needle-tip materials are arrays etched by template method, and their functional parts are limited by the small volume of the tip. The nano-tip copper synthesized by laser has high purity. Meanwhile, it can maximize and tune its electric field effect to regulate its catalytic performance, which is simple and convenient. This unique approach enables the achievement of exceptional selectivity and current density simultaneously for CO_2 and nitrate reduction, even in acidic electrolytes. Secondly, through the synergistic effects of the electric field and abundant interfaces, our catalyst exhibits accelerated CO_2RR and NITRR while effectively suppressing hydrogen evolution. This unique combination enables the

simultaneous achievement of high current densities and high selectivity for C₂₊ products and ammonia, surpassing the performance of most reported catalysts. Finally, by leveraging the dual effects of kinetics and thermodynamics enabled by the unique structure, we provide insights that can contribute to closing the carbon and nitrogen cycles simultaneously. We believe that this unique perspective based on two-pronged approach will broaden the horizon regarding the design of electrocatalysts and provide new insight into the mass production and the improvement of the corresponding industrial green chemical process.

There is no doubt that these three articles provide a good study of the role of copper oxidation state in CO₂RR; they focus on the relationship between the oxidation state of the Cu catalyst and the performance during the CO₂RR. In addition to the field effect, we also focus on verifying that the presence of abundant interfaces is beneficial to the adsorption of the post-dimerization intermediate (*OCCOH), thus reducing the energy barrier of C–C dimerization according to the result of in situ FTIR and online DEMS analysis and DFT calculation.

Meanwhile, since these three articles provide us with a thorough understanding of the role of copper oxidation state in CO₂RR, we have cited these three works on Page 3 in our revised manuscript as Ref. 30-32, which are highlighted in red color, as shown below.

“[30] Scholten, F., Sinev, I., Bernal, M. & Roldan Cuenya, B. Plasma-Modified Dendritic Cu Catalyst for CO₂ Electroreduction. *ACS Catal.* **9**, 5496-5502, (2019).

[31] Velasco-Vélez, J.-J. *et al.* The role of the copper oxidation state in the electrocatalytic reduction of CO₂ into valuable hydrocarbons. *ACS Sustain. Chem. Eng.* **7**, 1485-1492, (2018).

[32] Chou, T. C. *et al.* Controlling the oxidation state of the cu electrode and reaction intermediates for electrochemical CO₂ reduction to ethylene. *J. Am. Chem. Soc.* **142**, 2857-2867, (2020).”

We also appreciate the Reviewer for the thoughtful review and comments on our work. We have tried our best to revise our manuscript according to the valuable comments from the Reviewer.

Comment 2: 1: The quality of the SEM image in Fig.S2 is insufficient to discern the growth trajectory of the purported Cu_xO bipyramids, as asserted by the authors. It is suggested that TEM be employed to provide a more detailed view of the material's morphology at various time intervals.

Response: We thank the Reviewer very much for this very useful suggestion. Based on the suggestion of the reviewer, we have characterized the growth process of L-Cu_xO by TEM. The detailed TEM results are shown in **Supplementary Fig. 3**. According to the TEM results, firstly, the PLAL process produced dispersed small nanoparticles (5 min), and then these nanoparticles gradually grew up (10 min) under continuous laser stimulation. Next, the Cu_xO nanoparticles self-assembled into loosely interconnected agglomerates, and further merged under the action of the laser to form a bipyramidal structure (30 min), which continued to grow and mature, and eventually developed into a Cu_xO bipyramidal body (60 minutes).

We have also added **Supplementary Fig. 3** about the growth process of L-Cu_xO by TEM on Page 4 in our revised Supplementary Information, as shown below.

Supplementary Fig. 3 The growth process of L-Cu_xO-HC and its structural evolution during the synthetic process. TEM of L-Cu_xO-HC in different time during laser ablation.

Comment 3: 2. The approach of distinguishing between CuO and Cu₂O based on the brightness and darkness of the TEM image lacks rigorousness. It is advisable to employ Energy-Loss Near-Edge Structure (ELNES) analysis as a more reliable method for determining the valence state of 3d metals. Additionally, conducting Selected Area Electron Diffraction (SAED) is essential to substantiate the authors' conclusions.

Response: We thank the Reviewer a lot for raising this helpful suggestion. Based on the suggestion of the reviewer, we have added the electron energy loss spectroscopy analysis (**Supplementary Fig. 8**), selected area electron diffraction (SAED) and FFT patterns on different zones (**Supplementary Fig. 9**) to characterize the L-Cu_xO-HC material. According to the result of electron energy loss spectroscopy (EELS) spectra of O K-edge and Cu L-edge of L-Cu_xO-HC, the oxygen content in the dark area is higher than that in the bright area, and the atomic ratio of copper to oxygen is obtained by fitting the energy loss curve, which is close to CuO and Cu₂O, respectively. It is consistent with selected area electron diffraction (SAED) and the XRD result of L-Cu_xO-HC. (**Fig. 1h and Supplementary Fig. 9**).

We have added some descriptions on this point, which are highlighted in red color on Page 5 in our revised manuscript, as shown below.

“electron energy loss spectroscopy (EELS) spectra of O K-edge and Cu L-edge (Supplementary Fig. 8) reveals that the dark and bright domains are dominant by the CuO and Cu₂O phases, respectively, which agrees with selected area electron diffraction (SAED) and X-ray diffraction (XRD) pattern results (Fig. 1h and Supplementary Fig. 9).”

We have also added the **Supplementary Fig. 8** and **Supplementary Fig. 9** which are highlighted on Page 9 in our revised Supplementary Information, as shown below.

Supplementary Fig. 8. Structural characterization of L-Cu_xO-HC. (a,b) Spherical aberration -corrected high-resolution HAADF-STEM images and corresponding electron energy loss spectroscopy (EELS) spectra of O K-edge and Cu L-edge and fitted atomic percent of L-Cu_xO-HC.

Supplementary Fig. 9. Structural characterization of L-Cu_xO-HC. (a) Spherical aberration -corrected high-resolution HAADF-STEM images and corresponding FFT pattern and (b) selected area electron diffraction (SAED) of L-Cu_xO-HC.

Comment 4: 3. In the present manuscript, an array of characterization techniques, including TEM, SEM and XRD, were utilized to probe the structures and compositions of Cu_xO bipyramids. Notwithstanding, these characterizations were exclusively focused on the Cu_xO bipyramids prior to catalyzing CO₂ reduction, with post-reaction analyses notably absent. The author declared the angular configuration of the tips influenced the distribution of the electric field, but it has been previously documented that the morphology of certain Cu-based nanocatalysts can undergo substantial modifications under significant reduced current during CO₂RR. Consequently, comprehensive physicochemical characterizations subsequent to electrolysis are imperative to ascertain the stability of these nanostructures.

Response: We thank the Reviewer very much for providing this very important question. We have done many characterizations on our post-reaction catalysts, including XRD, XPS, TEM and SEM (**Supplementary Fig. 23-26**). In addition, we also systematically compared the EXAFS and EIS results on our catalysts before and after the reaction (**Supplementary Fig. 27 and 28**). According to the above characterization results, in terms of the morphology, the overall morphology of this catalyst is well preserved (SEM and TEM). In terms of the chemical state, although some metallic

Cu is observed, it is clearly shown that the dominant chemical state is still an oxidized state (XRD, EXAFS, EIS, XPS). We believe that the $\text{Cu}^{\delta+}$ contributes to the whole CO_2RR process as the performance of our catalysts always shows a better performance than the pure metallic Cu. Thus, it is complex copper oxidation state during CO_2RR that facilitates the electroreduction of carbon dioxide to ethylene and C_{2+} products, which is also consistent with other articles⁷⁻⁹.

Supplementary Fig. 23. Characterizations of the tested L-CuxO-HC. (a-e) TEM images and (f) SEM image of L-CuxO-HC after stability test. (Blue circle is the carbon black from GDL)

Supplementary Fig. 26. Characterizations of the tested L-CuxO-HC. (a) XRD pattern. (b) O 1s XPS spectra and (c) Cu 2p XPS spectra of L-CuxO-HC before and after stability test.

Supplementary Fig. 27. (a) Normalized XANES, (b) Fourier transformed EXAFS of L-Cu_xO-HC before and after CO₂RR.

Supplementary Fig. 28. EIS measurement before and after the stability test on L-Cu_xO-HC at -0.1 V vs. RHE in 1 M KOH electrolyte with a scanning frequency range from 1000 kHz to 0.1 Hz.

Comment 5: 4. The authors concluded that the Cu⁺/Cu²⁺ interface could offer more efficiently active sites to promote CO₂RR to C₂₊. Although the notion that the oxidized state of Cu may be reduced to Cu⁰ during CO₂RR remains a contentious issue, given the diverse testing conditions employed by different researchers. It is noteworthy that the authors conducted their tests under a reduction current of 900 mA cm⁻², which raises questions regarding the ability of Cu to retain its

valence state under such strenuous reducing conditions. Therefore, the execution of in situ EXAFS is recommended to ascertain the actual catalytic sites.

Response: We thank the Reviewer a lot for providing this very helpful suggestion. We agree that in situ spectroscopies could provide information on the chemical status of Cu during CO₂RR. As the in situ X-ray synchrotron is in pressing demand, we here use in situ XRD electrochemical test, which is complementary to in situ EXAFS, to characterize our materials. The result is shown below.

The results show that when a negative potential of -0.8 to -3.2 V vs RHE without iR-correction is applied, CuO, Cu₂O, and Cu are observed to coexist; these states can be maintained at high potentials. This is also consistent with the EXAFS, XPS, EIS results and XRD results showing copper oxide signal, similar to some articles¹⁰⁻¹². Therefore, we think the actual catalytic sites during CO₂RR are the coexistence of Cu (II), Cu (I), or Cu (0) (Cu^{δ+}). Post-analysis of the sample after electrolysis (**Supplementary Fig. 26 and 27**) shows that most metallic Cu will return to the oxidized state.

We have added **Supplementary Fig. 29**, which is highlighted in red color on Page 21, in our revised Supporting Information, as shown below.

Supplementary Fig. 29. X-ray diffraction (XRD) study in 1 M KOH electrolyte with flow cell.

A customized electrochemical cell was used with a Pt wire and an Ag/AgCl electrode as the counter and reference electrodes, respectively. The cathode and anode compartments were separated using an AEM membrane. The catalysts were dropped onto gas diffusion electrode as the working electrode. The electrolyte (1 M KOH) and CO₂ circulate through the cell during the measurements. The measurements were conducted at open circuit potential and from -0.8 V to -3.2 V vs. RHE without iR correction.

Comment 6: 5. It would be appreciative if the authors could elucidate the principles of testing in Fig. 3d, supplemented with pertinent literature citations for further context and validation.

Response: We thank the Reviewer very much for providing this very important suggestion. From a mathematical point of view, this principle of OH⁻ adsorption test is following the interaction energy of an adsorbate under an applied electric field, which can be written as follows¹³⁻¹⁵:

$$E_{ads} = E_0 + \mu\varepsilon - \frac{\alpha\varepsilon^2}{2}$$

where E_{ads} represents the adsorption energy at a given strength of electric field; E_0 is the adsorption energy in the absence of electric field; μ and α are the intrinsic dipole moment and polarizability values for the adsorbate, which is -0.26 eÅ and $0.14 \text{ eÅ}^2 \text{ V}^{-1}$ for OH⁻, respectively¹⁵; and ε refers to the electric field. It can be seen from the calculation that the stronger the electric field effect, the lower the adsorption energy of OH⁻, therefore OH⁻ could be adsorbed at a lower voltage under the influence of stronger electric field.

In addition, as Prof. Edward H. Sargent's work said¹⁰, "hydroxide ions on or near the copper surface could lower the CO₂ reduction and carbon monoxide (CO)-CO coupling activation energy barriers". Therefore, we conducted the OH⁻ adsorption test to characterize the ability of adsorb OH⁻ on L-Cu_xO-HC to evaluate its contribution to the good performance.

It is pronounced OH_{ad} peaks associated with Cu (100) facets on L-Cu_xO on the potential of ~0.4 V. Electrochemical adsorption of OH⁻ on the L-Cu_xO follows this formula: $\text{Cu}_x\text{O} + \text{OH}^- \rightarrow \text{Cu}_x\text{O}(\text{OH})_{ad} + \text{e}^-$. When a positive potential is applied, the electrode surface material will adsorb the anion OH⁻. Therefore, the higher the applied potential, the easier the adsorption of the OH⁻. If the OH⁻ adsorption reaction can occur at a relatively lower potential on certain materials, it means that

the material is more likely to adsorb OH^- compared to other materials, which is also consistent with the results of these work^{10,16}.

Based on the reviewer's suggestion, we have added more detailed discussion on **Supplementary Note 1** about OH^- adsorption test, which is highlighted in red color on Page 47 in our revised Supplementary Information. They are also shown below.

“From a mathematical point of view, this principle of OH^- adsorption test is following the interaction energy of an adsorbate under an applied electric field, which can be written as follows¹³⁻¹⁵:

$$E_{ads} = E_0 + \mu\varepsilon - \frac{\alpha\varepsilon^2}{2}$$

where E_{ads} represents the adsorption energy at a given strength of electric field; E_0 is the adsorption energy in the absence of electric field; μ and α are the intrinsic dipole moment and polarizability values for the adsorbate, which is -0.26 eÅ and $0.14 \text{ eÅ}^2 \text{ V}^{-1}$ for OH^- , respectively¹⁵; and ε refers to the electric field. It can be seen from the calculation that the stronger the electric field effect, the lower the adsorption energy of OH^- , therefore OH^- could be absorbed at a lower voltage under the influence of stronger electric field.

It is pronounced OH_{ad} peaks associated with Cu (100) facets on L- Cu_xO on the potential of $\sim 0.4 \text{ V}$. Electrochemical adsorption of OH^- on the L- Cu_xO follows this formula¹⁶: $\text{Cu}_x\text{O} + \text{OH}^- \rightarrow \text{Cu}_x\text{O}(\text{OH})_{ad} + \text{e}^-$. When a positive potential is applied, the electrode surface material will adsorb the anion OH^- . Thus, the higher the applied potential, the easier the adsorption of the OH^- . If the OH^- adsorption reaction can occur at a relatively lower potential on a certain material, it means that the material is more likely to adsorb OH^- compared to other materials^{10,16}.”

Comment 7: 6. Some basic electrochemical characterization needs to be provided, such as electrochemical impedance spectroscopy, electrochemical stability testing for at least 10 hours, partial current density of the product, etc.

Response: We thank the Reviewer a lot for this very important suggestion. The **Fig. 2a** provides the partial current density of all products. To further show the partial current of the product more

intuitively, we have added a new diagram about the partial current of the products (Supplementary Fig. 19). Meantime, we have added electrochemical impedance spectroscopy and electrochemical stability testing for 12 hours, as shown in Supplementary Fig. 20 and Fig. 2d on Page 19 in our revised Supplementary Information and Page 20 in our revised manuscript, respectively. They are also shown below.

Supplementary Fig. 19. Current density of each CO₂RR product and H₂ on L-Cu_xO-HC at different applied potentials in 1 M KOH (without iR-correction).

Supplementary Fig. 20. EIS measurement of L-Cu_xO-HC at different potentials in 1 M KOH electrolyte with a scanning frequency range from 1000 kHz to 0.1 Hz.

Fig. 2 CO₂ electroreduction performance. **a** Current density (top) and FE values (bottom) of each CO₂RR product and H₂ on L-Cu_xO-HC at different applied potentials in 1 M KOH (without iR-correction). **b** FE values of C₂₊ (top) and C₂H₄ yield production (bottom) for L-Cu_xO-HC, L-Cu_xO-MC, L-Cu_xO-LC, Cu-tip, C-CuO and C-Cu at their best performance. **c** C₂₊ current density of each catalyst at different applied potentials. **d** Stability measurement in 1 M KOH under a high current density of $\sim 600 \text{ mA cm}^{-2}$.

Reviewer #3

Comment 1: Herein, the authors employed a laser-assisted technique to synthesize Cu_xO bipyramids for electrochemical CO_2RR . The sharp tip angle of the nanostructure favors cation accumulation which stabilizes reaction intermediates via field effect and therefore assists the reaction kinetics. Also, the partially oxidized Cu interface further favors the formation of a critical intermediate for C-C coupling. Overall, the study is extensive, the hypothesis is well supported by various spectroscopic analyses and theoretical simulations. Furthermore, in addition to the fundamental scientific exploration, practical industrial application has also been demonstrated. I feel that the manuscript can be accepted in Nature Communications after the following minor concerns be addressed:

Response: We highly appreciate the Reviewer for the review and comments on the novelty of our work. We have tried our best to revise our manuscript according to the valuable comments from the Reviewer.

Comment 2: 1. In Supplementary Fig. 21, the authors presented XPS and PXRD analysis of the potential driven transformation of the catalyst during electrochemical CO_2RR . The data suggests that the surface is still a partially oxidized Cu_xO structure. However, does the metallic Cu phase appear during electrolysis? Enhancement of Cu-Cu scattering feature in EXAFS and shift of Cu-K edge in Xanes after electrolysis would be able to answer this question. Also, Is there any difference in impedance before and after long-term electrolysis?

Response: We thank the Reviewer a lot for raising up this important question. Based on the reviewer's suggestion, we acquired the EXAFS data and electrochemical impedance spectroscopy of L- Cu_xO -HC after long-term electrolysis, as shown in **Supplementary Fig. 27 and 28**. For the question "does the metallic Cu phase appear during electrolysis?", according to the comparative results of XRD, XPS, EIS and EXAFS on L- Cu_xO -HC before and after the reaction, some metallic Cu is observed, while the dominant species is still in the oxidized state. The XANES (Fig. S27a) shows that peak of L- Cu_xO -HC-AF at ~8995 eV shifts to lower energy and the pre-edge lies between C- Cu_2O and C- CuO , suggesting a reduction in oxidation state after electrolysis. EXAFS fitting (Fig. S27b) shows the minor contribution from metallic Cu-Cu after electrolysis.

We believe that during the reaction process, especially at high potentials, it comprises complex valence states such as $\text{Cu}^{0/+1/+2}$ interfaces that contribute to the high performance. This was supported by the in situ XRD (**Supplementary Fig. 29**). The catalytic performance of copper in a complex valence state is better than that of copper in a single valence state, which is also consistent with other articles⁷⁻⁹.

Based on the Reviewer's suggestion, we have added the **Supplementary Fig. 27 and 28** about EXAFS data and electrochemical impedance spectroscopy of L-Cu_xO-HC after long-term electrolysis, which are highlighted in red color on Page 24 in our revised Supplementary Information. They are also shown below.

Supplementary Fig. 27. (a) Normalized XANES, (b) Fourier transformed EXAFS of L-Cu_xO-HC before and after CO₂RR.

Supplementary Fig. 28. EIS measurement before and after stability in L-Cu_xO-HC taken at -0.1 V vs. RHE in 1 M KOH electrolyte with a scanning frequency range from 1000 kHz to 0.1 Hz.

Comment 3: 2. It would be interesting to see the CO₂RR trend with another alkali metal cation such as Li⁺. The interfacial accumulation of larger cations (K⁺, Cs⁺) is relatively higher than the smaller ones. However, does the electrostatic field at the tip curvature assist the accumulation of smaller cations as well? From a fundamental point of view, it would be interesting to observe if the tip-assisted accumulation diminishes the cation size effect.

Response: We thank the Reviewer very much for raising up this interesting suggestion. Based on the suggestion of the reviewer, we conducted the adsorption reactions of lithium ions and cesium ions separately on L-Cu_xO-HC (high electric field) and C-CuO (low electric field). According to the results of cation adsorption experiments, the strong electrostatic field at the tip curvature could increase the accumulation of different cations with 2-4 times. Meanwhile, the electrostatic field at the tip curvature exacerbates the gap in the size effect of the cations (**Supplementary Fig. 33**). Then we conducted their carbon dioxide catalytic performance in different cationic solutions to further evidence the result. However, in lithium hydroxide solution, due to severe flooding and hydrogen production, gas products could not be collected in our online testing system and there was almost no liquid products according to NMR result, so we only compared the catalytic performance with potassium ions and cesium ions. It is found that with potassium hydroxide or cesium hydroxide solutions, the FE of the product does not change a lot for L-Cu_xO-HC and C-CuO. In terms of current density, the current density of C-CuO in cesium hydroxide increased compared to that in potassium hydroxide, which might be due to the cationic effect. This mainly because that the corrected Stokes radius of hydrated cations show a decreasing trend from Li⁺ to K⁺ and Cs⁺, which is contrary to that of the ionic radius of cations (**Supplementary Table 1**). The difference in Stokes radius can lead to a varied change in the total amount of cations in the vicinity of the electrode and the electric field at the tip curvature will intensify the difference in the total amount of cation adsorption, resulting in different CO₂RR performances¹⁷. This is also well consistent with the results of those papers^{18,19}.

Supplementary Table 1. The radius of typical alkali metal cations in different states.

Cations	R_{ionic} (Å)	$R_{\text{ion-water}}$ (Å)	R_{Stokes} (Å)	R_{cStokes} (Å)
Li^+	0.76	2.08	2.38	3.82
K^+	1.50	2.79	1.25	3.31
Cs^+	1.91	3.13	1.19	3.29

Note: R_{ionic} represents the ionic radius of typical cations. $R_{\text{ion-water}}$ represents mean ion-water internuclear distances. R_{Stokes} and R_{cStokes} represent the Stokes and corrected Stokes radius of hydrated ions, respectively²⁰.

We have also added the results as **Supplementary Fig. 33, Supplementary Table 1** and the corresponding discussion, which are highlighted in red color on Pages 25 and 48 in our revised Supplementary Information, as shown below.

Supplementary Fig. 33. (a) Cation adsorption experiment and (b) FE of each product in electrolytes with different cations of L- Cu_xO -HC and C-CuO at their best potential; (c, e) FE of each product at different potentials and (d, f) the corresponding i-t curves of L- Cu_xO -HC and C-CuO in electrolytes with Cs^+ cations, respectively.

We conducted the adsorption reactions of lithium ions and cesium ions separately on L- Cu_xO -HC (high electric field) and C-CuO (low electric field). According to the results of cation adsorption experiments, the strong electrostatic field at the tip curvature could increase the accumulation of

different cations with 2-4 times, and meanwhile, the electrostatic field at the tip correlates with cations (Supplementary Fig. 33). Then we conducted their carbon dioxide catalytic performance in different solutions. However, in lithium hydroxide solution, due to severe flooding and hydrogen production, gas products could not be collected in the online testing system and there was almost no liquid product according to NMR result, so we only compared the catalytic performance with potassium and cesium ions. It is found that in potassium hydroxide or cesium hydroxide solutions, the FE of the product does not change much for L-Cu_xO-HC and C-CuO. In terms of current density, the current density of C-CuO in cesium hydroxide increases compared to that in potassium hydroxide, possibly due to the cationic effect. This is mainly because the corrected Stokes radius of hydrated cations show a decreasing trend from Li⁺ to K⁺ and Cs⁺, which is contrary to that of the ionic radius of cations (**Supplementary Table 1**). The difference in Stokes radius can lead to a varied change in the total amount of cations in the vicinity of the electrode and the electric field at the tip curvature will intensify the difference in the total amount of cation adsorption, resulting in different CO₂RR performances^{17-19,21}.

Supplementary Table 1. The radius of typical alkali metal cations in different states

Cations	R _{ionic} (Å)	R _{ion-water} (Å)	R _{Stokes} (Å)	R _{cStokes} (Å)
Li ⁺	0.76	2.08	2.38	3.82
K ⁺	1.50	2.79	1.25	3.31
Cs ⁺	1.91	3.13	1.19	3.29

Note: R_{ionic} represents the ionic radius of typical cations. R_{ion-water} represents mean ion-water internuclear distances. R_{Stokes} and R_{cStokes} represent the Stokes and corrected Stokes radius of hydrated ions, respectively²⁰.

Reference

- [1] Fang, J. Y. *et al.* Ampere-level current density ammonia electrochemical synthesis using CuCo nanosheets simulating nitrite reductase bifunctional nature. *Nat. Commun.* **13**, 7899, (2022).
- [2] Han, S. *et al.* Ultralow overpotential nitrate reduction to ammonia via a three-step relay mechanism. *Nat. Catal.* **6**, 402–414, (2023).
- [3] Gao, Q. *et al.* Breaking adsorption-energy scaling limitations of electrocatalytic nitrate reduction on intermetallic CuPd nanocubes by machine-learned insights. *Nat. Commun.* **13**, 2338, (2022).
- [4] Gao, Q. *et al.* Synthesis of core/shell nanocrystals with ordered intermetallic single-atom alloy layers for nitrate electroreduction to ammonia. *Nat. Synth.* **2**, 624–634, (2023).
- [5] Zhang, Q. & Asthagiri, A. Solvation effects on DFT predictions of ORR activity on metal surfaces. *Catal. Today.* **323**, 35-43, (2019).
- [6] Wang, Y. *et al.* Enhanced nitrate-to-ammonia activity on copper-nickel alloys via tuning of intermediate adsorption. *J. Am. Chem. Soc.* **142**, 5702-5708, (2020).
- [7] Zhang, W. *et al.* Atypical oxygen-bearing copper boosts ethylene selectivity toward electrocatalytic CO₂ reduction. *J. Am. Chem. Soc.* **142**, 11417-11427, (2020).
- [8] Chou, T. C. *et al.* Controlling the oxidation state of the Cu electrode and reaction intermediates for electrochemical CO₂ reduction to ethylene. *J. Am. Chem. Soc.* **142**, 2857-2867, (2020).
- [9] Zhang, J. *et al.* Grain Boundary-Derived Cu(+)/Cu(0) Interfaces in CuO Nanosheets for Low Overpotential Carbon Dioxide Electroreduction to Ethylene. *Adv Sci (Weinh).* **9**, 2200454, (2022).
- [10] Dinh, C. T. *et al.* CO₂ electroreduction to ethylene via hydroxide-mediated copper catalysis at an abrupt interface. *Science.* **360**, 783-787, (2018).
- [11] Li, H. *et al.* C₂₊ selectivity for CO₂ electroreduction on oxidized Cu-based catalysts. *J. Am. Chem. Soc.* **145**, 14335-14344, (2023).
- [12] Wang, P. *et al.* Boosting electrocatalytic CO₂-to-ethanol production via asymmetric C-C coupling. *Nat. Commun.* **13**, 3754, (2022).
- [13] Che, F. *et al.* Elucidating the roles of electric fields in catalysis: A perspective. *ACS Catal.* **8**, 5153-5174, (2018).
- [14] Chen, L. D., Urushihara, M., Chan, K. & Nørskov, J. K. Electric field effects in electrochemical CO₂ reduction. *ACS Catal.* **6**, 7133-7139, (2016).
- [15] Resasco, J. *et al.* Promoter effects of alkali metal cations on the electrochemical reduction of carbon dioxide. *J. Am. Chem. Soc.* **139**, 11277-11287, (2017).
- [16] Cao, Y. *et al.* Surface hydroxide promotes CO₂ electrolysis to ethylene in acidic conditions. *Nat. Commun.* **14**, 2387, (2023).
- [17] Yu, J. *et al.* Interfacial electric field effect on electrochemical carbon dioxide reduction reaction. *Chem Catalysis.* **2**, 2229-2252, (2022).
- [18] Gu, J. *et al.* Modulating electric field distribution by alkali cations for CO₂ electroreduction in strongly acidic medium. *Nat. Catal.* **5**, 268–276, (2022).
- [19] Ringe, S. *et al.* Understanding cation effects in electrochemical CO₂ reduction. *Energy Environ. Sci.* **12**, 3001-3014, (2019).
- [20] E. R. Nightingale, J. Phenomenological theory of ion solvation. effective radii of hydrated ions. *J. Phys. Chem. B.* **63**, 1381-1387, (1959).
- [21] Endrodi, B. *et al.* Operando cathode activation with alkali metal cations for high current density operation of water-fed zero-gap carbon dioxide electrolyzers. *Nat Energy.* **6**, 439-448, (2021).

REVIEWERS' COMMENTS

Reviewer #1 (Remarks to the Author):

The authors have addressed all my concerns and thus I would suggest the acceptance of this manuscript as is.

Reviewer #2 (Remarks to the Author):

The authors have carefully addressed the comments. I would recommend to accept this version.

Reviewer #3 (Remarks to the Author):

In the revised manuscript, the authors carefully addressed all of the concerns and provided appropriate justification. Their justifications are supported by experimental evidences and appropriate reference. Hence, this version of the manuscript can be accepted for publication in its current format.